# Association Between Early Point-of-Care Ultrasound and Emergency Department Outcomes in Admitted Patients with Non-Traumatic Abdominal Pain: A Propensity Score-Weighted Cohort Analysis

**DOI:** 10.3390/diagnostics15243182

**Published:** 2025-12-12

**Authors:** Meng-Feng Tsai, Fen-Wei Huang, Te-Fa Chiu, Tse-Chyuan Wong, Sheng-Yao Hung, Wei-Jun Lin, Shih-Hao Wu

**Affiliations:** 1Department of Emergency Medicine, China Medical University Hospital, Taichung 404327, Taiwan; apex0627@hotmail.com.tw (M.-F.T.); workforstatistics@gmail.com (F.-W.H.); tefachiu@gmail.com (T.-F.C.); tsechyuan@gmail.com (T.-C.W.); yao103022428@gmail.com (S.-Y.H.); linwejn@gmail.com (W.-J.L.); 2School of Medicine, College of Medicine, China Medical University, Taichung 404328, Taiwan; 3College of Public Health, China Medical University, Taichung 406040, Taiwan

**Keywords:** ultrasound, clinical approach, mortality, computed tomography, ED crowding

## Abstract

**Background**: To evaluate the association of point-of-care ultrasound (PoCUS) performed within one hour of emergency department (ED) arrival with ED length of stay (LOS) and healthcare costs in admitted ED patients with non-traumatic abdominal pain. **Methods:** This retrospective, inverse probability of treatment weighting (IPTW) cohort study was conducted at a tertiary medical center in Taiwan. This study analyzed data from 2021–2023, focusing on adult patients admitted to an ordinary ward with non-traumatic abdominal pain. Patients discharged from the ED, admitted to the ICU, or receiving PoCUS > 1 h (N = 864) were excluded. The final cohort of 6866 patients comprised those receiving PoCUS within 1 h (N = 1542) and those receiving no PoCUS (N = 5324). Primary and secondary outcomes (ED LOS, costs) were adjusted for age, gender, triage, vital signs, BMI, and comorbidities using generalized linear models with a Gamma distribution. **Results:** After IPTW adjustment in 6866 admitted abdominal pain patients, PoCUS within one hour was associated with a 14% shorter ED LOS (RM 0.86, 95% CI 0.83–0.89). A notable finding was that PoCUS performed within one hour was associated with 44% higher odds of CT utilization (OR 1.44, 95% CI 1.25–1.65) and 5% lower total healthcare costs (RM 0.95, 95% CI 0.91–0.99). Stratification by CT use revealed distinct patterns underlying these associations: in the non-CT subgroup, PoCUS was associated with 12% lower ED costs (RM 0.88, 95% CI 0.83–0.94), whereas in the CT subgroup, it was associated with 9% lower admission costs (RM 0.91, 95% CI 0.86–0.96). **Conclusions:** In admitted patients, PoCUS performed within one hour was associated with shorter ED LOS and lower total costs, despite a concurrent association with higher CT utilization. These findings are consistent with a dual, context-dependent role for PoCUS: associated with reduced ED costs in non-CT pathways and lower admission costs in CT pathways. However, as this is an observational study, these results represent associations rather than causal effects and may be influenced by unmeasured confounding. Prospective trials are required to validate these findings.

## 1. Introduction

Diagnostic decision-making for non-traumatic abdominal pain differs markedly between patients admitted to the hospital and those discharged directly from the emergency department (ED) [1,2]. For admitted patients, early clarification of the underlying condition is pivotal as it shapes subsequent processes, including specialty consultation, ward allocation, operative planning, and the timing of definitive therapy [3,4]. Delays during the initial ED evaluation can cascade throughout the inpatient course, underscoring the operational importance of timely assessment in this population [5].

Point-of-care ultrasound (PoCUS) has become an important bedside tool for the initial assessment of abdominal pain [6,7]. Performed by emergency physicians, PoCUS provides real-time visualization of intra-abdominal organs, vascular structures, and free fluid without the delays of patient transfer [8]. It has proven diagnostic value for conditions such as cholelithiasis, appendicitis [9], hydronephrosis, and aortic aneurysm [10], and its immediate availability can accelerate decision-making and improve ED throughput [11,12].

Computed tomography (CT) remains the gold standard for many abdominal emergencies due to its superior spatial resolution and comprehensive anatomical detail [4,13]. However, CT carries drawbacks, including radiation exposure, contrast-related risks, limited availability in crowded EDs, and higher costs [14,15,16]. Concerns about overuse have prompted efforts to refine diagnostic pathways and optimize imaging strategies [17].

Prior retrospective research on PoCUS has yielded mixed results often limited by confounding by indication. However, recent prospective randomized controlled trials [18,19] have overcome these limitations and consistently demonstrated that PoCUS integration significantly shortens ED length of stay (LOS) and reduces the need for additional imaging in patients with abdominal pain.

Despite this high-quality evidence, the literature has largely focused on the general ED population, leaving the specific impact of examination timing insufficiently characterized in the more complex cohort of patients requiring hospital admission. This study focuses specifically on patients admitted to an ordinary ward. This population represents a critical intermediate group for ED throughput. Unlike patients discharged from the ED (often with self-limiting conditions) or those requiring immediate ICU resuscitation, ward-admitted patients typically undergo extensive diagnostic workups where delays are common.

In this context, the conceptual framework for early PoCUS is its potential role as a pathway accelerator rather than solely a diagnostic substitute. It is hypothesized that PoCUS performed within the first hour may enhance early risk stratification, prompting swifter decisions for definitive imaging (such as CT) or specialist consultation. To rigorously evaluate this, a propensity score-weighted analysis was conducted to assess the association of PoCUS performed within one hour with ED LOS and healthcare costs, specifically accounting for the selection bias inherent in observational imaging studies.

## 2. Methods

### 2.1. Study Design and Setting

This study utilized a retrospective cohort design with propensity score weighting (IPTW). The analysis was conducted at a tertiary academic medical center in Taiwan with an annual census exceeding 160,000 ED visits. Data were retrieved from the electronic medical records (EMR) system for the period between 1 January 2021, and 31 December 2023. The Institutional Review Board approved the protocol (CMUH113-REC2-008) and waived the requirement for informed consent. The reporting of this study adheres to the STROBE statement for observational research [20].

### 2.2. Study Population

The study population was derived from a parent cohort of adult patients (≥18 years) presenting with a chief complaint of non-traumatic abdominal pain, as described in our previous work [12]. Explicitly, the current analysis represents a separate, hypothesis-driven secondary study with reference to the previous study, designed to specifically evaluate the admitted cohort. From this initial pool, patients were strictly screened to include only those admitted to an ordinary ward. Patients were excluded if they were (i) discharged from the ED, (ii) admitted directly to the intensive care unit (ICU), (iii) transferred to another facility, (iv) left against medical advice, or (v) died in the ED.

To evaluate the specific association of early PoCUS with outcomes, the admitted cohort was categorized based on the timing of the ultrasound examination: (1) no PoCUS and (2) PoCUS within 1 h of ED arrival. Patients who received PoCUS more than 1 h after arrival were excluded from the primary analysis to isolate the potential efficiency gains of early assessment; however, this excluded subgroup was subsequently analyzed in a sensitivity analysis to assess selection bias.

We applied propensity score-weighting to compare the PoCUS within 1 h group (N = 1542) directly against the No PoCUS group (N = 5324) (Figure 1).

#### 2.2.1. PoCUS Practice and Training

Details regarding the institutional sonography training curriculum and quality assurance protocols have been published previously [12]. Briefly, examinations were performed by emergency medicine residents and attending physicians who had completed standardized training requirements certified by the Taiwan Society of Ultrasound in Medicine. Image quality and interpretation were monitored through regular audit meetings to ensure operator proficiency and minimize variability.

#### 2.2.2. Imaging Modality Selection

Diagnostic imaging (PoCUS, CT, or both) was selected by the treating physician based on clinical judgment, considering patient acuity, suspected pathology, and departmental workflow. PoCUS functioned as an extension of the physical examination, typically performed immediately after the primary survey to guide subsequent testing (e.g., laboratories or CT). Its use was pragmatic, influenced by operator proficiency, confidence, and departmental crowding.

For frequency, although serial PoCUS examinations may be used for monitoring in practice, this study analyzed only the index (first) PoCUS performed during the ED visit to assess its association with initial disposition and efficiency outcomes.

### 2.3. Variables

Demographic data, triage acuity (Taiwan Triage and Acuity Scale, TTAS) [21], vital signs, comorbidities, and ICD-10-CM diagnostic codes were extracted from the EMR. Key operational timepoints, including arrival, imaging ordering/completion, and ED departure, were recorded.

### 2.4. Outcome Measures

The primary outcome was ED LOS at the index visit, defined as the interval from triage registration to ED departure for ward admission. ED LOS was selected as the primary endpoint to reflect operational efficiency and ED throughput, which are critical metrics in the context of departmental crowding. Secondary outcomes included healthcare costs (ED costs, admission costs, and total costs) calculated for the index episode of care, encompassing the ED visit and the subsequent inpatient admission. While cost structures vary by health system, these metrics were included to evaluate the relative economic implications of resource utilization pathways.

### 2.5. Data Analyses

#### 2.5.1. Descriptive Statistics

Quantitative variables were described using means ± standard deviations (SD) or medians (interquartile range, IQR), depending on normality. Differences were assessed via Student’s *t*-tests or Mann–Whitney U tests. Qualitative variables were presented as counts and proportions, with comparisons conducted using the Chi-square test.

#### 2.5.2. Missing Data

Data incompleteness was negligible (<1% across all key variables) and primarily restricted to anthropometric measures and initial vital signs. Since this study explicitly excluded patients admitted to the ICU or those who died in the ED, the cohort comprised exclusively of patients deemed stable enough for ward admission. Consequently, missing vital signs in this group are attributable to random clerical omissions rather than clinical instability (i.e., patients too unstable for measurement). Given the low frequency and non-informative nature of these missing values in this specific population, a complete case analysis approach was adopted without imputation.

#### 2.5.3. Propensity Score Weighting

To address potential confounding by indication, inverse probability of treatment weighting (IPTW) was employed [22]. A propensity score for receiving early PoCUS was estimated using a multivariable logistic regression model incorporating age, sex, triage level, vital signs, BMI, and comorbidities. The goodness-of-fit for this model was confirmed using the Hosmer–Lemeshow test (*p* = 0.605). Stabilized weights were applied to the cohort. Weight distribution diagnostics indicated a range from a minimum of 1.00 to a maximum of 14.20. Extreme weights were rare, with only 0.16% (n = 11) of the cohort having a weight exceeding 10. The effective sample size after weighting was 4447. Given this negligible proportion of extreme weights and the preserved sample size, weight trimming was not performed; instead, a doubly robust approach was utilized to mitigate any potential influence from outliers while retaining the full study population. Covariate balance was assessed using standardized mean differences (SMDs), with a threshold of <0.1 indicating adequate balance.

#### 2.5.4. Sensitivity and Subgroup Analyses

To assess potential selection bias resulting from the exclusion of patients receiving delayed ultrasound, a sensitivity analysis was performed. A separate IPTW model was constructed to compare outcomes between the excluded PoCUS > 1 h cohort and the No PoCUS cohort.

Additionally, a targeted subgroup analysis was conducted for patients with Taiwan Triage and Acuity Scale (TTAS) levels 3 and 4. This subgroup was selected to isolate the population with the highest diagnostic uncertainty, excluding critical patients (levels 1–2), who typically follow standardized resuscitation protocols or require urgent CT to exclude life-threatening pathology, and non-urgent patients (level 5), who rarely require extensive imaging.

Finally, a pre-specified exploratory stratification based on CT usage (With CT vs. Without CT) was performed to independently evaluate the association of PoCUS with resource utilization in these two clinically distinct diagnostic pathways. Interaction terms (PoCUS × CT Use) were included in the models to assess the statistical validity of subgroup differences. Since these analyses were exploratory, strict multiplicity adjustments were not applied; findings should be interpreted as hypothesis-generating.

#### 2.5.5. Outcome Analysis

Outcomes were compared between the weighted groups using Generalized Linear Models (GLMs) with a Gamma distribution and log link function. Model goodness-of-fit was evaluated using the scaled deviance divided by degrees of freedom; a ratio of approximately 1.0 (specifically 1.097 in our primary model) indicated adequate fit. To minimize the impact of extreme propensity scores, stabilized weights were employed. Furthermore, a doubly robust approach was adopted by adjusting the weighted GLMs for the same covariates used in the propensity score model (age, sex, triage level, vital signs, BMI, and comorbidities).

Results are reported as Ratios of Means (RMs) or Odds Ratios (ORs) with 95% confidence intervals (CIs). An RM represents the proportional change in the outcome; for example, an RM of 0.86 implies a 14% reduction in the mean value. To facilitate clinical interpretation, absolute differences calculated using average marginal effects are also presented alongside RMs. Finally, to quantify the potential impact of unmeasured confounding on the primary outcome, the E-value was calculated [23,24]. This metric estimates the minimum strength of association that an unmeasured confounder would need to have with both the exposure and the outcome to explain away the observed association, conditional on the measured covariates.

All analyses were performed using SAS software (version 9.4; SAS Institute Inc., Cary, NC, USA), with statistical significance defined as a two-sided *p*-value < 0.05.

## 3. Results

### 3.1. Study Population and Characteristics

6866 index ED visits were entered in the core analyses (Figure 1). Of them, 5324 (77.5%) were in the No-PoCUS group and 1542 (22.5%) in the PoCUS within 1 h group.

The baseline characteristics of the study population before and after IPTW are detailed in Table 1. Before IPTW, the groups showed significant baseline differences. The PoCUS within 1 h group was younger, had a higher median BMI, included more patients with triage level 3, and had fewer comorbidities. The PoCUS group also had a higher rate of CT utilization. After IPTW, these baseline covariates were well-balanced, with most Standardized Mean Differences (SMDs) falling below 0.1, indicating negligible imbalance.

### 3.2. Primary and Secondary Outcomes (IPTW-Adjusted)

Adjusted outcomes for the IPTW cohort are presented in Table 2.

#### 3.2.1. Primary Outcome (ED LOS)

For the primary outcome, PoCUS performed within one hour was associated with a significant 14% reduction in ED LOS (Ratio of Means [RM] 0.86, 95% CI 0.83–0.89, *p* < 0.001). This corresponds to an adjusted absolute reduction of approximately 2.2 h.

#### 3.2.2. Secondary Outcomes (Costs)

Patients in the PoCUS group had 44% higher odds of receiving a CT scan in the ED (Odds Ratio [OR] 1.44, 95% CI 1.25–1.65, *p* < 0.001). While ED costs were not significantly different (RM 1.03, 95% CI 1.00–1.05, *p* = 0.059), the PoCUS group was associated with a 6% reduction in subsequent admission costs (RM 0.94, *p* = 0.009; absolute reduction: -NT$ 5235) and total costs (RM 0.95, *p* = 0.016; absolute reduction: -NT$ 5015).

### 3.3. Sensitivity and Subgroup Analyses

To ensure robustness and explore clinical heterogeneity, a series of additional analyses were conducted. First, a sensitivity analysis regarding the excluded cohort (N = 864) was performed to assess selection bias. Subsequently, exploratory subgroup analyses (stratified by Triage level and CT usage) were conducted to investigate potential effect modification and resource utilization patterns across distinct clinical pathways.

#### 3.3.1. Sensitivity Analysis of Excluded Cohort

Analysis of the excluded PoCUS > 1 h cohort (N = 864) versus the No PoCUS group (N = 5324) revealed a different pattern. After IPTW adjustment, delayed PoCUS was associated with an 11% increase in ED LOS (RM 1.11, 95% CI 1.07–1.15, corresponding to +1.77 h, *p* < 0.001), contrasting with the 14% reduction (RM 0.86, 95% CI 0.83–0.91, corresponding to −2.2 h, *p* < 0.001) observed with early PoCUS (Table 2 vs. Appendix A). This opposite effect confirms that the observed efficiency gain is specifically time-dependent rather than due to patient selection for PoCUS. Outcomes for this cohort are detailed in Appendix A.

#### 3.3.2. Targeted Subgroup Analysis: Triage Levels 3 & 4

As shown in Table 3, in the subgroup of patients with triage levels 3 and 4, the association with reduced ED LOS remained strong (RM 0.87, 95% CI 0.83–0.90, *p* < 0.001). This group also showed significant reductions in admission costs (RM 0.88, *p* < 0.001) and total costs (RM 0.90, *p* < 0.001).

#### 3.3.3. Exploratory Stratified Analysis by CT Use

When stratified by CT use (Table 4), the reduction in ED LOS was significant in both patients without CT (RM 0.80, 95% CI 0.74–0.86, *p* < 0.001) and patients with CT (RM 0.87, 95% CI 0.84–0.91, *p* < 0.001). In the without-CT group, PoCUS was associated with a 12% reduction in ED costs (RM 0.88, *p* < 0.001), but no significant change in admission or total costs. In the with-CT group, PoCUS showed no significant change in ED costs but was associated with a 9% reduction in admission costs (RM 0.91, *p* < 0.001) and an 8% reduction in total costs (RM 0.92, *p* < 0.001).

#### 3.3.4. Exploratory Subgroup Analysis by Diagnostic Category

Finally, to identify potential heterogeneity in treatment associations across different pathologies, an exploratory subgroup analysis was performed across 18 diagnostic categories (Figure 2). As shown in the forest plot, the association with reduced LOS was consistent across most high-prevalence groups. However, estimates for lower-prevalence conditions (e.g., aortic dissection) had wider confidence intervals due to smaller sample sizes, warranting cautious interpretation.

The benefit of PoCUS is most significant in accelerating the diagnostic and disposition process for high-urgency and ultrasound-sensitive diseases. This is seen in conditions like bowel perforation and peritonitis (RM 0.58, 95% CI 0.46 to 0.74) and disorders of the gallbladder and biliary tract (RM 0.74, 95% CI 0.69 to 0.81).

For the largest single group, Nonspecific abdominal pain (N = 3346), PoCUS still provided a 14% reduction in ED LOS (RM 0.86, 95% CI 0.82 to 0.90), an association consistent with the overall cohort’s benefit.

However, for conditions highly reliant on CT for diagnosis, such as diseases of the appendix (RM 1.04, 95% CI 0.97 to 1.11) and dissection or aneurysm (RM 0.98, 95% CI 0.57 to 1.69), PoCUS did not shorten ED LOS. Furthermore, in clinically complex scenarios, PoCUS was associated with a significant prolongation of ED LOS. This was seen in “pregnancy, childbirth and the puerperium” (RM 2.09, 95% CI 1.65 to 2.65) and thromboembolism (RM 2.24, 95% CI 1.14 to 4.41).

## 4. Discussions

In this propensity score-weighted (IPTW) cohort study of 6866 admitted abdominal pain patients, several key findings emerged. We confirmed that PoCUS within one hour was associated with a 14% lower ED LOS. The central finding was a paradoxical one: early PoCUS was linked to 44% higher CT utilization yet 5% lower total healthcare costs. Stratification analysis suggested that this pattern is driven by distinct clinical pathways: PoCUS was associated with lower ED costs in patients without CT, and with lower admission costs in patients with CT. Furthermore, subgroup analysis showed these benefits were not uniform, being most significant for ultrasound-sensitive conditions like bowel perforation and biliary tract disease, but absent in others, such as appendicitis.

One of the most notable and seemingly paradoxical findings of this study is that early PoCUS was associated with 44% higher odds of CT utilization. This result appears to contradict numerous studies advocating for PoCUS as a tool to reduce CT scans and radiation exposure [25,26]. However, this counter-intuitive finding must be interpreted within the specific clinical context of this cohort: “non-traumatic abdominal pain patients requiring hospital admission”.

In many clinical settings, and particularly in Taiwan, a CT scan is often considered the standard diagnostic procedure before or upon admission for abdominal pain patients whose conditions are complex enough to warrant hospitalization [11,27]. This is done to maximize diagnostic certainty and avoid medical errors [4,28]. In this “CT-as-routine” pathway, the role of PoCUS naturally shifts from a substitute for CT to a complement to it. Therefore, the higher CT use associated with PoCUS may suggest that sonographic findings (e.g., free fluid, free air, inflammatory change, or hydronephrosis) served as markers of complexity, identifying patients who warranted further definitive evaluation, rather than PoCUS acting as a standalone diagnostic endpoint.

The association of early PoCUS with both increased CT use and lower total costs presents a complex dynamic. The exploratory stratification by CT use (Table 4) reveals patterns consistent with a hypothesized dual role for PoCUS across different clinical pathways, although it must be emphasized that these mechanisms are inferred from utilization data rather than directly observed:

First, in the subgroup of patients who ultimately did not receive a CT, the data align with a Substitute model. In this group, early PoCUS was associated with a significant 12% reduction in ED costs (RM 0.88, *p* < 0.001). While this association is susceptible to selection bias (where PoCUS is preferentially selected for diagnostically simpler cases), quantifying this pathway remains clinically relevant. It highlights a distinct pattern of resource utilization where PoCUS use aligns with diagnostic parsimony and lower costs.

Second, in the subgroup of patients who did receive a CT, the data support a Complementary or Pathway Acceleration hypothesis. In this group, PoCUS had no significant association with ED costs (*p* = 0.408), which is logical as the expensive CT scan was still performed. However, early PoCUS was associated with a significant 9% reduction in admission costs (RM 0.91, *p* < 0.001) and an 8% reduction in total costs (RM 0.92, *p* < 0.001). Even when performed alongside CT, PoCUS contributes dynamic information (e.g., organ function, sonographic tenderness) not available from static imaging. While the exact mechanism cannot be confirmed retrospectively, this pattern is consistent with a scenario where positive PoCUS findings validate the need for admission and expedite the subsequent workup, thereby reducing downstream costs despite the initial imaging investment.

Notably, the association with reduced LOS was most pronounced in the Triage 3 and 4 subgroup. This effect modification aligns with clinical expectations: unlike critical patients (Triage 1–2), whose pathway is typically dictated by immediate resuscitation or urgent CT to exclude life-threatening pathology, and non-urgent patients (Triage 5), the Triage 3–4 cohort represents the population with the highest diagnostic ambiguity. In this intermediate group, PoCUS likely exerts its greatest utility by facilitating earlier risk stratification and accelerating disposition decisions.

Regarding the validity of the study cohort, the sensitivity analysis provides robust evidence against pure selection bias: if physicians simply chose PoCUS for diagnostically easier patients, delayed PoCUS should have shown similar benefits. Instead, delayed PoCUS was associated with prolonged LOS (+1.77 h), suggesting that timing itself is the critical factor. This time-dependent dose–response pattern (early PoCUS = shorter LOS; delayed PoCUS = longer LOS; no PoCUS = intermediate) is more consistent with a genuine efficiency effect than with confounding by indication alone.

An alternative explanation warrants consideration. In the “With CT” cohort, early PoCUS may not reduce admission costs through complementary diagnostic insights. Instead, its primary role may be as a “CT acceleration tool” or “diagnostic pathway accelerator.” In this framework, an early concerning PoCUS finding, such as free fluid, hydronephrosis, or a dilated aorta, could prompt expedited CT ordering and prioritization. This acceleration of the diagnostic pathway would facilitate earlier specialist involvement and timelier inpatient management, with downstream cost reductions as a secondary effect. Current observational data cannot definitively distinguish between these two mechanisms (substitution versus acceleration) due to the lack of granular data on physician intent and decision-making timing. Resolving this ambiguity would require a prospective randomized controlled trial where patients with diagnostic uncertainty are randomized to PoCUS or standard care prior to the decision for advanced imaging.

The exploratory subgroup analysis (Figure 2) identifies variations in the association between early PoCUS and ED LOS across different pathologies. These variations appear to follow three distinct patterns:

1. Acceleration (The “Rule-Out” Pattern): For the largest cohort, “Nonspecific abdominal pain” (N = 3346), PoCUS was associated with a 14% reduction in ED LOS (RM 0.86). This benefit was particularly pronounced in ultrasound-sensitive conditions such as biliary tract disease (RM 0.74). This finding aligns with recent disease-specific evidence demonstrating that early PoCUS significantly shortens ED LOS and reduces costs for patients with biliary pathology, even among those requiring admission after an initial discharge [29]. This reinforces the utility of PoCUS in pathways where sonographic findings are highly specific and directive. This association is consistent with PoCUS functioning as a “rule-out” tool, potentially providing sufficient diagnostic confidence to accelerate disposition in clinically ambiguous patients.

2. Neutrality (The “CT-Reliant” Pattern): PoCUS was not associated with shorter LOS for conditions mandating CT, specifically “Diseases of the appendix” (RM 1.04) and “Dissection or aneurysm” (RM 0.98). In patients eventually diagnosed with appendicitis, PoCUS was likely utilized during the initial assessment of undifferentiated symptoms (e.g., epigastric or periumbilical pain) to screen for pathology or rule out mimics such as ureteral stones causing hydronephrosis. However, since confirmatory CT remains the standard for surgical planning or definitive exclusion [30,31], PoCUS cannot bypass this rate-limiting step. Similarly, for the stable, ward-admitted aortic cohort (N = 51), CT is essential for anatomical definition [32,33]. In these pathways, the workflow is dictated by advanced imaging, specialist consultation, and medical management (e.g., blood pressure control), rendering the initial PoCUS time-neutral.

3. Deceleration (The “Risk-Stratification” Tool): Furthermore, in several clinically complex scenarios, PoCUS was associated with a significant prolongation of ED LOS. This was seen in “Thromboembolism” (RM 2.24) and “Pregnancy, childbirth and the puerperium” (RM 2.09). This finding should not be interpreted as a PoCUS-induced inefficiency. Rather, it aligns with the clinical demands of these high-risk pathologies. For example, in pregnancy, the primary objective of early PoCUS is often risk stratification (e.g., ruling out ectopic pregnancy) [34]. Identifying a patient as high risk or indeterminate typically necessitates specialist consultation and extended observation, inherently prolonging ED LOS. Similarly, for thromboembolism, the pathway is dictated by the initiation of anticoagulation and admission planning [35,36]. Therefore, while outcome data are required to confirm safety benefits, the observed prolongation in LOS is consistent with a shift toward more intensive, guideline-concordant care pathways rather than diagnostic delay.

The observed economic outcomes present a complex dynamic. While ED costs showed no significant difference (*p* = 0.059), Total Costs were significantly reduced by 5% (*p* = 0.016). The lack of strong significance in ED costs likely reflects a cancellation effect: the additional expense of increased CT utilization (OR 1.44) in the PoCUS group was likely neutralized by the savings associated with shorter ED LOS. Specifically, reduced ED dwell time inherently decreases the consumption of time-dependent resources, such as nursing surveillance, intravenous fluid maintenance, and repeated medication dosing, thereby mitigating the upfront cost of advanced imaging.

However, the significant reduction in admission costs suggests that the value of early PoCUS extends beyond the ED. By potentially identifying pathologies earlier or clarifying clinical trajectories, early PoCUS may enable more focused inpatient management, resulting in downstream savings. From a healthcare systems perspective, a 5% reduction in total per-patient expenditure represents a clinically meaningful improvement in resource stewardship. However, this benefit presents a complex trade-off regarding radiation exposure.

Unlike prior studies focused on specific conditions like nephrolithiasis or small bowel obstruction, where PoCUS reduced radiation [24,25], our study analyzed a broader cohort of undifferentiated abdominal pain. This population inherently includes life-threatening pathologies—such as ischemic bowel or aortic dissection—where CT remains the essential gold standard for diagnosis. Consequently, the association with higher CT utilization implies an increased aggregate radiation burden.

Nevertheless, this finding must be contextualized within the contemporary clinical environment. In an era where diagnostic precision and patient safety are paramount, CT has become the de facto standard for evaluating adult patients admitted with abdominal pain [4,28]. The increased utilization likely reflects a safety-first strategy where PoCUS facilitates the rapid decision to obtain definitive imaging. While reducing radiation remains a goal, in this complex admitted cohort, the priority appears to be the accurate exclusion of lethal pathology, for which the combination of PoCUS and CT is increasingly utilized.

### Strengths and Limitations

This study examined a broad spectrum of non-traumatic abdominal pain presentations using a large sample, extended timeframe, and robust statistical adjustments. Several limitations warrant consideration.

First, and most fundamentally, the strict cohort selection criteria (excluding ED discharges and ICU admissions) restrict the scope of this study to inpatient workup efficiency. It should be noted that the current analysis represents a separate, hypothesis-driven secondary study with reference to the previous study, intentionally isolating this subgroup to verify outcome consistency in the specific context of ward admission. This design inherently prevents any evaluation of PoCUS as a tool to avoid hospitalization or modify disposition decisions. Consequently, the findings apply strictly to the specific workflow of stable patients already destined for ward admission.

Second, regarding unmeasured confounding, variables such as physician gestalt, pain severity, and momentary ED crowding were not captured. To quantify the potential impact of this bias, the E-value for the primary outcome (RM 0.86) was calculated. The E-value for the point estimate was 1.60. This implies that an unmeasured confounder (e.g., physician proficiency or specific clinical severity) would need to be associated with both early PoCUS exposure and the outcome of shorter LOS by a risk ratio of at least 1.60 to fully explain away the observed association. While it is plausible that unmeasured factors exist, they would need to exert a substantial and simultaneous influence to completely nullify the findings.

Specific domains of unmeasured confounders warrant consideration:
(i)Physician factors: Variables such as operator experience and individual preference for ultrasound were not captured. It is plausible that physicians who proactively perform early PoCUS possess other attributes of efficiency (e.g., faster decision-making style). Thus, early PoCUS may serve partly as a marker of a more expedited care trajectory, rather than solely a direct mediator of it. This potential confounding by physician efficiency could bias the results in favor of the PoCUS group.(ii)Clinical and systemic factors: Granular details (e.g., pain location, physical exam findings) and systemic pressures (e.g., ED crowding, CT scanner availability) were not explicitly modeled. Importantly, however, the direction of bias introduced by these factors may not necessarily inflate the PoCUS benefit. For example, if PoCUS were preferentially used in easier patients (selection bias), one would expect lower CT utilization. In contrast, the observation of higher CT use in the PoCUS group suggests that these patients are likely to represent a more diagnostically complex cohort. Thus, unmeasured confounding may have masked the full extent of PoCUS-related efficiency gains, rendering the cost-saving estimates conservative.

Third, the distinct clinical profiles of the two groups (e.g., age and comorbidities) necessitated large weights to achieve balance, resulting in increased variance (wide standard deviations) in the post-IPTW cohort. While stabilized weights and doubly robust adjustment were used to mitigate this, the limited overlap suggests that the findings are most applicable to the range of patients where clinical equipoise exists. The substantial reweighting required to achieve covariate balance (effective N = 4447 out of 6866) indicates limited natural overlap between groups. However, the time-dependent pattern observed (early PoCUS = benefit; delayed PoCUS = harm) supports a genuine effect beyond patient selection.

Fourth, the exclusion of the PoCUS > 1 h cohort (N = 864) introduces potential selection bias, as these patients may represent a distinct clinical phenotype with evolving presentations. Critically, our sensitivity analysis (Appendix A) demonstrated that delayed PoCUS was associated with an 11% increase in ED LOS (RM 1.11, +1.77 h), directly opposite to the 14% reduction observed with early PoCUS. This reversal of effect direction provides strong evidence that the observed benefit is genuinely time-dependent. If selection bias alone explained our findings (i.e., PoCUS was used more often in simpler patients), both early and delayed PoCUS should show similar benefits. The fact that delayed PoCUS shows harm suggests that the timing of PoCUS is mechanistically important, partially addressing concerns about unmeasured confounding.

Fifth, this single-center study reflects a specific institutional context characterized by standardized PoCUS residency training, high-volume workflows, and readily available 24 h CT access. Consequently, these results may not be generalizable to settings with different credentialing standards, lower sonographic proficiency, or limited access to advanced imaging.

Sixth, several operational and data-related limitations exist.
(i)Data & Population: Missing data were minimal (<1%) and handled via complete case analysis. Findings are limited to adults and cannot be extrapolated to pediatric populations.(ii)Granularity & Quality: Clinical diagnoses and PoCUS indications were not standardized, reflecting real-world variability. Due to the dataset structure, specific sonographic findings prompting CT ordering could not be analyzed. Furthermore, as a retrospective review, real-time validation of image quality and inter-rater reliability were not feasible; accuracy relied on clinician judgment consistent with routine practice.(iii)Operational & Temporal Factors: ED LOS may have been shaped by unmeasured factors such as ward availability. Additionally, the study period (2021–2023) encompasses the COVID-19 pandemic. While both groups operated under shared environmental constraints, the unique operational pressures of this era may limit generalizability.

Finally, regarding outcome measures, the study was specifically designed to evaluate operational efficiency (LOS and costs) rather than clinical prognosis. Consequently, long-term outcomes such as in-hospital mortality or ICU transfer after admission were outside the scope of this analysis. Additionally, patient-centered measures (e.g., satisfaction, pain scores) were not captured, and standard return visit metrics are not applicable to this fully admitted cohort. Furthermore, the cost analysis is based on data from Taiwan’s National Health Insurance; therefore, absolute cost savings may not directly extrapolate to other healthcare systems, although relative efficiency trends are likely transferable

## 5. Conclusions

In this single-center, propensity score-weighted cohort study restricted to patients admitted to an ordinary ward, PoCUS performed within one hour was associated with shorter ED LOS and lower total healthcare cost. The data reveal patterns consistent with a hypothesized context-dependent role: PoCUS usage aligned with reduced ED costs in non-CT pathways and lower admission costs in CT pathways. However, these findings represent associations rather than causal effects and must be interpreted with caution due to the significant risk of confounding by indication (where physicians may select PoCUS for specific patient presentations). Therefore, these results should be viewed as hypothesis-generating. Prospective randomized controlled trials are essential to validate these efficiency gains and confirm safety before broadly altering clinical practice.

## Figures and Tables

**Figure 1 diagnostics-15-03182-f001:**
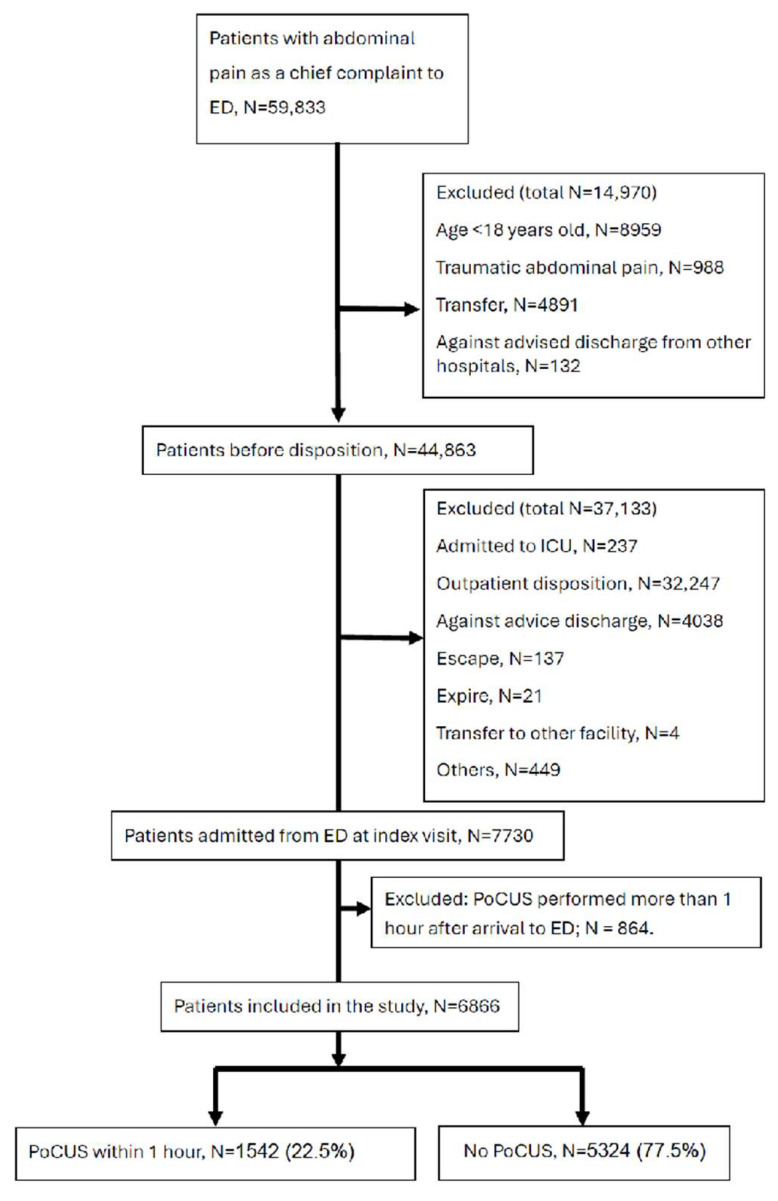
Flow diagram of study population selection. Comparison groups were defined based on the timing of the initial ultrasound. Note: The excluded PoCUS > 1 h cohort (N = 864) was subjected to a sensitivity analysis to assess potential selection bias. As detailed in Appendix A, this excluded group showed no significant difference in outcomes compared to the No PoCUS group, indicating that their exclusion did not systematically skew the primary findings.

**Figure 2 diagnostics-15-03182-f002:**
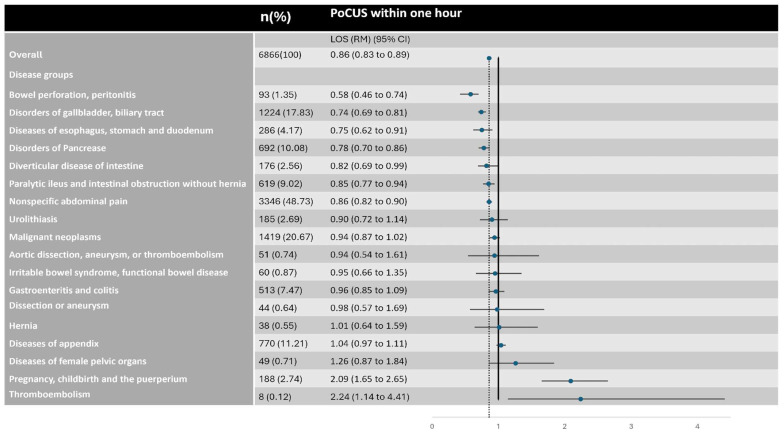
Exploratory subgroup analysis of ED LOS with PoCUS within one hour versus no PoCUS. Forest plot displays Ratios of Means (RM) with 95% Confidence Intervals. The sample size (n) for each subgroup is listed in the left-hand column. The solid vertical line represents the null effect (Ratio of Means = 1.0). The dotted vertical line indicates the overall estimate for the entire cohort (Ratio of Means = 0.86). Note: These analyses are exploratory in nature; *p*-values and confidence intervals have not been adjusted for multiplicity. Estimates for subgroups with small sample sizes (e.g., aortic dissection) have wide confidence intervals and should be interpreted with caution.

**Table 1 diagnostics-15-03182-t001:** Demographics of the study population at the index visit.

	Data Before IPTW			Data After IPTW		
	No-PoCUS N = 5324 (77.54)	PoCUS Within 1 h N = 1542 (22.46)	*p*	No-PoCUS	PoCUS Within 1 h	SMD
Age	58.00 (42.00 to 71.00)	51.00 (37.00 to 66.00)	<0.001	57.00 (41.00 to 70.00)	57.00 (41.00 to 70.00)	0.001
Sex			0.011			
M	2678 (50.30)	719 (46.63)		49.57	50.65	0.030
F	2646 (49.70)	823 (53.37)		50.43	49.35	−0.030
BMI	23.32 (20.66 to 26.40)	23.79 (20.96 to 26.99)	<0.001	23.31 (20.63 to 26.42)	23.81 (21.05 to 26.91)	−0.005
Triage			<0.001			
1	183 (3.44)	19 (1.23)		2.94	2.95	0.005
2	1361 (25.56)	288 (18.68)		24.01	23.96	−0.003
3	3740 (70.25)	1223 (79.31)		72.29	72.35	0.001
4	39 (0.73)	12 (0.78)		0.74	0.74	0.000
5	1 (0.02)	0 (0.00)		0.02	0.00	−0.284
HR	94.00 (81.00 to 109.0)	89.00 (76.00 to 105.0)		94.00 (81.00 to 109.0)	89.00 (76.00 to 105.0)	−0.129
SBP	127.0 (112.0 to 147.0)	129.0 (115.0 to 149.0)		127.0 (112.0 to 146.0)	131.0 (115.0 to 152.0)	0.090
DBP	78.00 (70.00 to 86.00)	80.00 (71.00 to 88.00)		78.00 (70.00 to 86.00)	79.00 (71.00 to 88.00)	0.044
BT	36.80 (36.30 to 37.40)	36.70 (36.30 to 37.20)		36.80 (36.30 to 37.40)	36.70 (36.20 to 37.20)	−0.087
RR	20.00 (19.00 to 20.00)	20.00 (18.00 to 20.00)		20.00 (19.00 to 20.00)	20.00 (18.00 to 20.00)	−0.063
Comorbidities	3896 (73.18)	929 (60.25)	<0.001	70.25	70.11	−0.002
ED resource utilization						
LOS in 1st ED (h)	11.10 (6.40 to 20.80)	9.40 (6.10 to 15.70)	<0.001	11.00 (6.30 to 20.60)	9.70 (6.20 to 16.50)	−0.107
1st ED cost (NTD)	11,610.5 (8346.5 to 14,645.5)	11,561.5 (8348.0 to 14,104.0)	0.125	11,514.0 (8281.0 to 14,460.0)	11,843.0 (8658.0 to 14,606.0)	0.0213
1st CT	3778 (70.96)	1185 (76.85)	<0.001	71.20	77.92	0.128

Data are presented as frequency (percentage). Continuous variables are presented as Median (Interquartile Range, IQR) to account for the non-normal distribution of weighted data. IPTW: inverse probability of treatment weighting; SMD: Standardized mean difference, with values < 0.1 indicating negligible imbalance and 0.1–0.2 suggesting minimal imbalance; PoCUS: point of care ultrasonography; CT: CT only; LOS: length of stay; ED: emergency department; NTD: New Taiwan Dollars.

**Table 2 diagnostics-15-03182-t002:** Adjusted Outcomes in the study population.

		Data Before IPTW		Data After IPTW		
Outcome Measures, RM or OR (95% CI)	No-Pocus	PoCUS-Within 1 h	*p*-Value	PoCUS-Within 1 h	Absolute Difference	*p*-Value
CT in ED	Ref.	1.34 (1.17 to 1.53)	<0.001	1.44 (1.25 to 1.65)		<0.001
ED LOS (h)	Ref.	0.87 (0.83 to 0.91)	<0.001	0.86 (0.83 to 0.89)	−2.2 h	<0.001
ED costs	Ref.	1.03 (0.99 to 1.06)	0.128	1.03 (1.00 to 1.05)	+392 NTD	0.059
Admission costs	Ref.	0.95 (0.89 to 1.00)	0.071	0.94 (0.90 to 0.99)	−5235 NTD	0.009
Total costs (total ED+ admission)	Ref.	0.96 (0.91 to 1.01)	0.094	0.95 (0.91 to 0.99)	−5015 NTD	0.016

Adjusted for age, gender, triage, BMI, and comorbidities. Data are presented as Gamma distribution with log link (handles right-skewed data). RM: ratio of means; OR: odds ratio; IPTW: inverse probability of treatment weighting; PoCUS: point of care ultrasonography; NTD: New Taiwan dollars.

**Table 3 diagnostics-15-03182-t003:** Targeted Subgroup Analysis of Adjusted Outcomes in Patients with Triage Levels 3 and 4.

		Data Before IPTW		Data After IPTW	
Outcome Measures, RM (95% CI)	No-Pocus	PoCUS-Within 1 h	*p*-Value	PoCUS-Within 1 h	*p*-Value
ED LOS (h	Ref.	0.87 (0.83 to 0.92)	<0.001	0.87 (0.83 to 0.90)	<0.001
ED costs	Ref.	1.02 (0.99 to 1.06)	0.215	1.03 (1.00 to 1.06)	0.094
Admission costs	Ref.	0.89 (0.83 to 0.94)	<0.001	0.88 (0.83 to 0.92)	<0.001
Total costs (total ED+ admission)	Ref.	0.90 (0.86 to 0.95)	<0.001	0.90 (0.86 to 0.94)	<0.001

Adjusted for age, gender, triage, BMI, and comorbidities. Data are presented as Gamma distribution with log link (handles right-skewed data). RM: ratio of means; IPTW: inverse probability of treatment weighting; PoCUS: point of care ultrasonography. Note: This analysis restricts the cohort to patients with Taiwan Triage and Acuity Scale (TTAS) levels 3 and 4. This subgroup was selected to isolate the population with the highest diagnostic uncertainty, distinguishing them from critical patients (levels 1–2) requiring immediate resuscitation or protocolized CT, and non-urgent patients (level 5) who typically require minimal investigation.

**Table 4 diagnostics-15-03182-t004:** Exploratory Stratified Analysis of Adjusted Outcomes by CT Utilization.

		Data Before IPTW		Data After IPTW	
Outcome Measures, RM (95% CI)	No-Pocus	PoCUS-Within 1 h	*p*-Value	PoCUS-Within 1 h	*p*-Value
Without CT					
ED LOS (h	Ref.	0.82 (0.74 to 0.91)	<0.001	0.80 (0.74 to 0.86)	<0.001
ED costs	Ref.	0.90 (0.83 to 0.98)	0.011	0.88 (0.83 to 0.94)	<0.001
Admission costs	Ref.	1.03 (0.91 to 1.16)	0.653	1.02 (0.93 to 1.11)	0.738
Total costs (total ED+ admission)	Ref.	1.01 (0.90 to 1.13)	0.845	1.00 (0.92 to 1.09)	0.980
With CT					
ED LOS (h)	Ref.	0.88 (0.84 to 0.93)	<0.001	0.87 (0.84 to 0.91)	<0.001
ED costs	Ref.	1.02 (0.99 to 1.05)	0.291	1.01 (0.99 to 1.04)	0.408
Admission costs	Ref.	0.91 (0.86 to 0.98)	0.007	0.91 (0.86 to 0.96)	<0.001
Total costs (total ED+ admission)	Ref.	0.93 (0.88 to 0.98)	0.011	0.92 (0.88 to 0.97)	<0.001

Adjusted for age, gender, triage, BMI, and comorbidities. Data are presented as Gamma distribution with log link (handles right-skewed data). RM: ratio of means; IPTW: inverse probability of treatment weighting; PoCUS: point of care ultrasonography. Note: Interaction testing was performed to assess effect modification by CT usage. The interaction term (PoCUS × CT Use) was statistically significant for ED LOS (*p* = 0.008), ED Costs (*p* < 0.001), and Admission Costs (*p* = 0.025), confirming that the association between PoCUS and these outcomes differs significantly between the strata.

## Data Availability

All relevant materials are presented in the present manuscript. The raw data supporting the conclusions of this article will be made available by the author, Shih-Hao Wu, without undue reservation. E-mail address: ambertwu@gmail.com.

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
