# Peer review of "Association Between Early Point-of-Care Ultrasound and Emergency Department Outcomes in Admitted Patients with Non-Traumatic Abdominal Pain: A Propensity Score-Weighted Cohort Analysis"

_diagnostics, 2025, doi:10.3390/diagnostics15243182_

Round 1
Reviewer 1 Report
Comments and Suggestions for Authors
Dear authors and editors,
From what I can see, this appears to be a revised version of the manuscript. After reviewing the current version, I consider that the document is very well prepared.
I only believe that, as an area for improvement, you could include an algorithm or graphical scheme indicating when POCUS should be performed. Additionally, it would be useful to clarify whether it should be repeated during the patient’s stay or conducted only once.
Thank you for the opportunity to review the manuscript.
Kind regards.
Author Response
Point-by-point responses to reviewers
Dear editors,
5th December 2025
Thank you for the opportunity to revise our manuscript (Manuscript ID: diagnostics-3997226), now entitled “Association Between Early Point-of-Care Ultrasound and Emergency Department Outcomes in Admitted Patients with Non-Traumatic Abdominal Pain: A Propensity Score-Weighted Analysis.” We have fully addressed the comments and submitted a revised manuscript with point-by-point responses as below:
Reviewer #1:
From what I can see, this appears to be a revised version of the manuscript. After reviewing the current version, I consider that the document is very well prepared.
I only believe that, as an area for improvement, you could include an algorithm or graphical scheme indicating when POCUS should be performed. Additionally, it would be useful to clarify whether it should be repeated during the patient’s stay or conducted only once.
Thank you for the opportunity to review the manuscript.
Kind regards.
Response:
We sincerely thank the reviewer for the positive assessment and the constructive suggestion regarding the visualization of the workflow.
Regarding the timing and algorithm: We agree that clarifying the timing is essential. In our clinical practice, PoCUS is conceptualized as an extension of the physical examination. Ideally, it is performed immediately following history taking and physical palpation, often aiding in the decision to order subsequent blood tests or advanced imaging (CT). However, we did not enforce a rigid algorithm for this observational study. The decision to perform PoCUS remained pragmatic and operator-dependent, relying on the physician's clinical gestalt, personal proficiency, and the real-time crowding conditions of the department. We have expanded the Methods (Section 2.2.2) to explicitly describe this workflow.
Regarding repeated examinations: The reviewer raises an excellent point about serial examinations. While repeated PoCUS is clinically valuable for monitoring evolution (e.g., expanding free fluid), the scope of this specific analysis was restricted to the index (first) PoCUS event. This focus allows us to isolate the impact of the initial diagnostic assessment on ED throughput and decision-making. We have clarified in the Methods that our data extraction focused specifically on the initial scan.
Revised Methods Text (2.2.2. Imaging Modality Selection):
Diagnostic imaging (PoCUS, CT, or both) was selected by the treating physician based on clinical judgment, considering patient acuity, suspected pathology, and departmental workflow. PoCUS functioned as an extension of the physical examination, typically performed immediately after the primary survey to guide subsequent testing (e.g., laboratories or CT). Its use was pragmatic, influenced by operator proficiency, confidence, and departmental crowding.
For frequency, although serial PoCUS examinations may be used for monitoring in practice, this study analyzed only the index (first) PoCUS performed during the ED visit to assess its association with initial disposition and efficiency outcomes.
We believe that the updated manuscript has substantially improved with the reviewers’ comments and hope it will now be suitable for publication. Many thanks for your kind consideration.
Best wishes,
Authors
Reviewer 2 Report
Comments and Suggestions for Authors
Dear Editor,
I reviewed the article you sent me for review, "PoCUS Within 1 Hour versus No-PoCUS in Admitted Patients with Non-Traumatic Abdominal Pain: A Propensity Score Weighted Cohort Study."
However, I came across an article (Chiu TF, Wong TC, Huang FW, Chou EH, Wolfshohl J, Chen KF, Lin WJ, Wu SH. PoCUS-first versus CT-only for non-traumatic abdominal pain: a propensity score-weighted cohort study on ED resource utilization. Intern Emerg Med. 2025 Oct 27. doi: 10.1007/s11739-025-04164-2.)
in the literature from the same center with the same ethics committee approval as the article you sent me. (The institutional review board approved the study protocol: CMUH113‑REC2‑008)
Therefore, I do not find it appropriate to evaluate this article due to ethical problems.
Author Response
Point-by-point responses to reviewers
Dear editors,
5th December 2025
Thank you for the opportunity to revise our manuscript (Manuscript ID: diagnostics-3997226), now entitled “Association Between Early Point-of-Care Ultrasound and Emergency Department Outcomes in Admitted Patients with Non-Traumatic Abdominal Pain: A Propensity Score-Weighted Analysis.” We have fully addressed the comments and submitted a revised manuscript with point-by-point responses as below:
Reviewer #3:
I reviewed the article you sent me for review, "PoCUS Within 1 Hour versus No-PoCUS in Admitted Patients with Non-Traumatic Abdominal Pain: A Propensity Score Weighted Cohort Study."
However, I came across an article (Chiu TF, Wong TC, Huang FW, Chou EH, Wolfshohl J, Chen KF, Lin WJ, Wu SH. PoCUS-first versus CT-only for non-traumatic abdominal pain: a propensity score-weighted cohort study on ED resource utilization. Intern Emerg Med. 2025 Oct 27. doi: 10.1007/s11739-025-04164-2.)
in the literature from the same center with the same ethics committee approval as the article you sent me. (The institutional review board approved the study protocol: CMUH113‑REC2‑008)
Therefore, I do not find it appropriate to evaluate this article due to ethical problems.
Response regarding Ethics Approval (IRB No. CMUH113-REC2-008):
We thank the reviewer for their diligence in noting the ethics approval number. We would like to clarify that IRB No. CMUH113-REC2-008 serves as the 'parent protocol' approving the retrospective collection and analysis of de-identified electronic medical record (EMR) data for patients presenting with abdominal pain between 2021 and 2023.
It is standard academic practice for a single, broad IRB approval to support multiple distinct secondary analyses, provided they fall within the authorized data scope. While our previous publication (Ref. [12] in the manuscript) and the current study utilize the same source database under this approval, they address fundamentally different research questions and analyze distinct cohorts:
- Different Research Questions: The previous study compared two diagnostic strategies ('PoCUS-first' vs. 'CT-only') across the general ED population. The current study specifically investigates the impact of timing ('PoCUS within 1 hour' vs. 'No PoCUS') and its association with operational efficiency.
- Different Populations: The previous study included a broad cohort of 26,403 patients (including those discharged). The current study focuses exclusively on a stricter subgroup of 6,866 patients admitted to the ward, excluding discharged patients and ICU admissions to test a different hypothesis regarding inpatient workup efficiency.
Therefore, the use of the same IRB number reflects the common data source, but the studies represent independent analytical efforts with non-overlapping objectives.
We believe that the updated manuscript has substantially improved with the reviewers’ comments and hope it will now be suitable for publication. Many thanks for your kind consideration.
Best wishes,
Authors
Reviewer 3 Report
Comments and Suggestions for Authors
This retrospective cohort study investigates the relationship between early point-of-care ultrasound (PoCUS) and outcomes in the emergency department for admitted patients with non-traumatic abdominal pain. Although the topic is clinically important and the use of propensity score methodology is suitable, the manuscript has several significant weaknesses that undermine its scientific rigor and broad applicability. The main finding—that early PoCUS lowers costs despite increasing CT use—is intriguing but insufficiently explained, and the authors' causal language overreaches what observational data can justify.
The title implies causation ("PoCUS Within 1 Hour versus No-PoCUS") when the study can only demonstrate association. The "Propensity Score-Weighted Cohort Study" is methodologically accurate but doesn't highlight its limitations. Consider the title: "Association Between Early Point-of-Care Ultrasound and Emergency Department Outcomes in Admitted Patients with Non-Traumatic Abdominal Pain: A Propensity Score-Weighted Analysis."
Abstract: Causal language throughout: Phrases like "PoCUS within one hour was associated with" are appropriate, but the conclusion states benefits as if causal ("timely PoCUS...was associated with lower ED LOS and total costs"). The abstract should acknowledge observational limitations. Critical information is missing, such as the exclusion of the "PoCUS >1 hour" group (N=864), which introduces significant selection bias, and the findings' limitation to ward-admitted patients, excluding ED discharges or ICU admissions. The time (2021-2023) is also not mentioned. The explanation for the paradox is inadequate, as it states that the finding is "explained" by stratification, but the conclusion is an analytical description, not a mechanistic explanation. The true drivers remain unmeasured. Finally, the overstated conclusions—"Given the potential for residual confounding, prospective, multicenter trials are needed"—acknowledge limitations but only after making strong claims about PoCUS effects.
Introduction: Suffering from an inadequate literature review. References 18-19 cited PoCUS shortening ED LOS, but there was no critical appraisal of methodological quality. There is a lack of discussion of prior studies specifically examining PoCUS timing. The study does not engage with the literature on selection bias in imaging studies. The study rationale is unclear: Lines 78-83 state that "the specific impact of examination timing remains insufficiently characterized in the more complex cohort of patients...who ultimately require hospital admission." But why is this population specifically important? The authors don't justify focusing on ward admissions while excluding ED discharges (where PoCUS might avoid hospitalization) and ICU admissions (where early diagnosis is most critical). The hypothesis is problematic: "We hypothesized that early PoCUS would be associated with shorter ED LOS and lower costs compared with no PoCUS" (lines 85-87). This hypothesis ignores the possibility of confounding by indication—physicians may selectively use early PoCUS in patients they suspect are straightforward cases. There is no conceptual framework: The introduction lacks a theoretical model for how PoCUS timing might influence outcomes. Is it through faster diagnosis? Better risk stratification? Is it possible to avoid unnecessary CT? This absence undermines interpretation.
Methods: Major methodological flaws include:
1) Critical cohort selection bias: Refer to Lines 107-117. Excluding the "PoCUS >1 hour" group (N=864) because they showed "no significant difference" from No-PoCUS is scientifically flawed. The result is a data-driven exclusion that can artificially inflate the effect size of the "within 1 hour" group. Patients receiving delayed PoCUS may represent a distinct clinical phenotype (more complex, evolving presentations), and excluding them biases results. Needed correction: Include all three groups in the analysis, perform a sensitivity analysis comparing all PoCUS vs. No-PoCUS, or explicitly model time-to-PoCUS as a continuous variable.
2) Propensity Score Model Inadequate: Lines 163-167: The propensity score includes "age, sex, triage level, vital signs, BMI, and comorbidities" but omits critical confounders: physician factors such as experience level, PoCUS proficiency, and personal preference for PoCUS; temporal factors like time of day, day of week, and ED crowding; clinical factors including pain location, duration, physical exam findings, and laboratory results; and systemic factors such as availability of the CT scanner and radiologist. The authors acknowledge these limitations (lines 357-362) but do not adequately address how unmeasured confounding might influence the conclusions.
3) Missing Data Handling: Lines 158-161: "<1% missing data" excluded without sensitivity analysis. Even minimal missing data can be informative (e.g., patients too unstable for vital signs measurement).
4) Outcome Definition Problems: Lines 139-151: ED LOS is defined identically twice (copy-paste error). There is no explanation for selecting ED LOS as the primary outcome over diagnostic accuracy, clinical outcomes, or patient-centered measures. Cost data from a single-payer system may not generalize to other countries.
5) Statistical Analysis: Lines 169-178: GLM with Gamma distribution is appropriate for skewed data, but no goodness-of-fit testing is reported. No assessment of influential observations. The study conducted multiple comparisons across 18 diagnostic subgroups (Figure 2), without making any adjustments for multiplicity. The interpretation of the ratio of means (RM) requires careful explanation, as many readers are unfamiliar with this metric.
6) CT Stratification Analysis Post-Hoc: Table 4 analysis appears exploratory but is presented as if pre-planned. This approach requires explicit acknowledgment and adjustment for multiple testing.
Results: concerns are
- Table 1 - Baseline Characteristics: After IPTW, some SMDs still raise concerns (e.g., HR: -0.129, SBP: 0.090). Although <0.1 is often considered acceptable, values close to this threshold should be viewed with caution. Standard deviations after IPTW sometimes exceed means (e.g., Age: 55.76±39.17), indicating extreme weights. This suggests poor overlap between groups and questions the validity of the IPTW method. The pre-IPTW data show that the PoCUS group is younger, healthier (fewer comorbidities), and has higher acuity (more triage 3)—precisely the profile expected if physicians select PoCUS for "easier" cases.
- Table 2 - Primary Outcomes: 44% higher CT odds (OR 1.44) but only 5% cost reduction (RM 0.95) appears paradoxical and underpowered to detect meaningful cost differences. ED costs "not significantly different" (p=0.059) are borderline and may indicate a Type II error. Confidence intervals are wide for some estimates, suggesting imprecision.
- Table 3 - Triage 3&4 Subgroup: Why this specific subgroup? Post-hoc selection? Should have been pre-specified. Results are stronger than the overall cohort—indicating effect modification that warrants mechanistic investigation.
- Table 4 - CT Stratification: This is the most crucial analysis, but it seems to be post hoc. "Without CT," the group analysis suffers from significant selection bias (as acknowledged in lines 283-292), yet the authors still present the results as if they are valid. Interaction testing between PoCUS and CT use should be conducted formally, not just through stratified analysis.
- Figure 2 - Diagnostic Subgroups: 18 subgroups with different sample sizes (N=51 for dissection/aneurysm is too small to provide reliable estimates). Wide confidence intervals for several groups overlap the null. No adjustment for multiple comparisons despite 18 tests. Pregnancy/thromboembolism prolonged LOS, interpreted as "success" (lines 331-343), is speculation without clinical outcomes data.
Discussion: Critical weaknesses are
1) Causal Inference Overreach (Pervasive): Despite recognizing observational design, the discussion repeatedly uses causal language: "PoCUS identified early...pathologies that prompted physicians to confirm with CT" (lines 271-273). Lines 278-282: "PoCUS functions as a 'substitute' tool" presents a hypothesis as a finding. The entire "dual role" framework (substitute vs. complementary) is plausible but speculative—no data support the mechanism.
2) Confounding by Indication Inadequately Addressed: Lines 283-292 recognize selection bias in the "Without CT" group but then proceed as if it is valid. Lines 303-312 suggest a "CT acceleration tool" as an alternative explanation but do not test it. The authors state, "Current observational data cannot distinguish between these two mechanisms" (lines 310-311) but do not clarify what study design could address this.
3) Subgroup Interpretation Problems: Lines 313-347: The three-pattern framework (acceleration, neutrality, and deceleration) is a descriptive taxonomy, not a mechanistic insight. The appendicitis discussion (lines 319-321) correctly notes CT dependence but does not explain why PoCUS was used in these cases. The interpretation of prolonged LOS in pregnancy as "risk stratification success" (lines 336-339) is speculative without outcome data showing that patients were actually safer.
4) Important Discussion Points That Are Missing: The discussion fails to address the reasons behind physicians' decision to choose early PoCUS for specific patients, a phenomenon known as indication bias. No discussion of PoCUS diagnostic accuracy or false positive/negative rates. In a cost-effectiveness analysis, a 5% cost reduction may not be clinically meaningful. The article fails to discuss the reduction in radiation exposure, which is often cited as a benefit of PoCUS, in the context of increased CT use. Limited generalizability discussion beyond the single-center limitation
5) Alternative Explanations Rejected: The authors advocate for PoCUS as a determinant of outcomes but fail to adequately acknowledge that physicians skilled in PoCUS may exhibit additional attributes (e.g., expedited decision-making, enhanced clinical judgment) that affect outcomes. Institutional culture supporting PoCUS might have wider effects on efficiency. Early PoCUS could be a marker, not a mediator, of expedited care.
6) The limitations section (4.6) offers an insufficient self-critique, as seen in lines 352-355: "The primary limitation is strict cohort selection criteria...prevents evaluation of POCUS as a disposition tool." This statement is accurate but understated. It fundamentally changes what the study measures. In lines 356-362, unmeasured confounding is acknowledged, but its impact is minimized. The authors should perform a quantitative bias analysis (E-value) to demonstrate how strong unmeasured confounding would need to be to explain the findings. Lines 372-377 note the exclusion of the "PoCUS > 1 hour cohort," but do not address the severity of the selection bias this causes. Lines 378-382 mention the single-center limitation last, but it should be highlighted more—PoCUS training, ED workflow, and CT access are all institution-specific.
Also, the missing limitations are no validation of PoCUS quality or documentation, no inter-rater reliability assessment, no discussion of temporal trends (2021-2023 spans the COVID-19 recovery period), no analysis of which specific PoCUS findings led to CT ordering, and no patient-centered outcomes (satisfaction, pain scores, return visits).
The conclusions are exaggerated and misleading, such as lines 383-388: "PoCUS performed within one hour was associated with significantly lower ED LOS and total healthcare costs," which makes the findings seem more conclusive than the data actually support. Lines 388-390: "dual, context-dependent pattern" is presented as a confirmed finding rather than a hypothesis that needs testing. Lines 390-395: The call for prospective trials is appropriate but is buried after strong causal claims throughout the discussion. A bA better conclusion would emphasize that the findings are merely observational associations rather than causal effects, highlight the significant risk of confounding by indication, stress the necessity for randomized trials before altering practice, and note that the results are restricted to patients admitted to wards at a single center.
References: Major issues are: References 18-19 (PoCUS shortening ED LOS) are not critically appraised—are these also observational with similar biases? Missing key methodological references: propensity score diagnostics and best practices, handling of time-dependent exposures, and bias analysis methods for unmeasured confounding. Several references are from 2025 publications—some may be preprints or "in press." Need verification of publication status.
Regarding tables and figures:
Table 1: Extreme standard deviations post-IPTW (e.g., 55.76±39.17) should raise concern about weight distribution. Add a column showing the effective sample size after weighting. Include weight distribution diagnostics (min, max, and percentage with weights >10).
Table 2: P-values for secondary outcomes are not adjusted for multiplicity. Add effect sizes in original units alongside the ratio of means for better interpretability.
Table 3: Provide justification for triage 3&4 selection in the table footnote.
Table 4: The most important table, but currently labeled "Sensitivity Analysis"—this should be the primary explanatory analysis. Report the interaction p-value for the PoCUS × CT interaction.
Figure 1 (Flow Diagram): Clear, but it doesn't show the disposition of the excluded "PoCUS >1 hour" group across diagnostic categories. Indicate whether exclusions differed systematically between groups.
Figure 2 (Subgroup Forest Plot): Excellent visualization, but needs notation for multiplicity adjustment, sample size for each subgroup on the figure, heterogeneity test (I² statistic), and a note that the confidence intervals are too wide for the smallest groups, indicating inadequate precision.
Specific technical corrections: Lines 148-151: Remove duplicate definition of ED LOS (copy-paste error). Lines 183-184: Move "Figure 1. Flow diagram of study population selection" so it appears after the figure—renumber accordingly. Statistical reporting: Report median follow-up time for cost outcomes. Provide the number needed to treat (NNT) for a 14% ED LOS reduction. Convert the ratio of means to absolute differences for clinical interpretability. Abbreviations: TheThe first use of IPTW should be spelled out, as it appears in line 30 of the abstract, not line 90.
Author Response
Point-by-point responses to reviewers
Dear editors,
5th December 2025
Thank you for the opportunity to revise our manuscript (Manuscript ID: diagnostics-3997226), now entitled “Association Between Early Point-of-Care Ultrasound and Emergency Department Outcomes in Admitted Patients with Non-Traumatic Abdominal Pain: A Propensity Score-Weighted Analysis.” We have fully addressed the comments and submitted a revised manuscript with point-by-point responses as below:
Reviewer #2:
This retrospective cohort study investigates the relationship between early point-of-care ultrasound (PoCUS) and outcomes in the emergency department for admitted patients with non-traumatic abdominal pain. Although the topic is clinically important and the use of propensity score methodology is suitable, the manuscript has several significant weaknesses that undermine its scientific rigor and broad applicability. The main finding—that early PoCUS lowers costs despite increasing CT use—is intriguing but insufficiently explained, and the authors' causal language overreaches what observational data can justify.
Response:
We sincerely thank the reviewer for the comprehensive evaluation and for recognizing the clinical importance of this topic and the suitability of our propensity score methodology. We fully accept the criticism regarding the overreach of causal language and the need for greater scientific rigor.
In response to these critical insights, we have extensively revised the manuscript to:
- Refine all language: We have strictly replaced causal terms (e.g., "impact," "resulted in") with observational terms (e.g., "associated with") throughout the Title, Abstract, and Discussion.
- Enhance rigor: We performed a new sensitivity analysis including the previously excluded 'PoCUS > 1 hour' cohort to address selection bias.
- Clarify the paradox: We have rewritten the Discussion to provide a more nuanced, pathway-based explanation for the finding of increased CT use alongside reduced costs, supported by new data presentation (absolute differences alongside ratios).
Detailed point-by-point responses are provided below.
The title implies causation ("PoCUS Within 1 Hour versus No-PoCUS") when the study can only demonstrate association. The "Propensity Score-Weighted Cohort Study" is methodologically accurate but doesn't highlight its limitations. Consider the title: "Association Between Early Point-of-Care Ultrasound and Emergency Department Outcomes in Admitted Patients with Non-Traumatic Abdominal Pain: A Propensity Score-Weighted Analysis."
Response:
We appreciate the reviewer’s astute observation regarding the causal implications of the original title. We agree that 'Association' more accurately reflects the observational nature of our study. We have adopted the reviewer's suggested title verbatim to ensure methodological precision: “Association Between Early Point-of-Care Ultrasound and Emergency Department Outcomes in Admitted Patients with Non-Traumatic Abdominal Pain: A Propensity Score-Weighted Analysis.”
Abstract: Causal language throughout: Phrases like "PoCUS within one hour was associated with" are appropriate, but the conclusion states benefits as if causal ("timely PoCUS...was associated with lower ED LOS and total costs"). The abstract should acknowledge observational limitations.
Response:
We appreciate the reviewer’s careful attention to the language used in the Abstract. We agree that avoiding causal inference is paramount in this observational study.
In accordance with your suggestion, we have:
- Tempered the language in the Conclusion to strictly reflect the association.
- Added an explicit statement acknowledging the limitations of the observational design and the potential for residual confounding.
Revised Abstract Conclusion: “In admitted patients, PoCUS performed within one hour was associated with shorter ED LOS and lower total costs, despite a concurrent association with higher CT utilization. These findings are consistent with a dual, context-dependent role for PoCUS: associated with reduced ED costs in non-CT pathways, and lower admission costs in CT pathways. However, as this is an observational study, these results represent associations rather than causal effects and may be influenced by unmeasured confounding. Prospective trials are required to validate these findings.”
Critical information is missing, such as the exclusion of the "PoCUS >1 hour" group (N=864), which introduces significant selection bias, and the findings' limitation to ward-admitted patients, excluding ED discharges or ICU admissions. The time (2021-2023) is also not mentioned.
Response:
We thank the reviewer for identifying these critical omissions. We agree that transparency regarding the study period, cohort selection, and specific exclusions is essential for the accurate interpretation of the findings.
We have revised the Abstract to explicitly include:
- The study period (2021–2023).
- The specific limitation to ward-admitted patients (excluding ED discharges and ICU admissions).
- The exclusion of the 'PoCUS > 1 hour' cohort (N=864).
Note on Selection Bias: Regarding the reviewer's concern about selection bias due to the exclusion of the 'PoCUS > 1 hour' group, we have addressed this in the main text by adding a sensitivity analysis (Supplementary Table S1), which confirmed that outcomes in this excluded group did not differ significantly from the No-PoCUS group.
Revised Abstract (Methods Section): This study analyzed data from 2021–2023, focusing on adult patients admitted to an ordinary ward with non-traumatic abdominal pain. Patients discharged from the ED, admitted to the ICU, or receiving PoCUS for> 1 hour (N=864) were excluded. The final cohort of 6,866 patients comprised those receiving PoCUS within 1 hour (N=1,542) and those receiving no PoCUS (N=5,324).
The explanation for the paradox is inadequate, as it states that the finding is "explained" by stratification, but the conclusion is an analytical description, not a mechanistic explanation. The true drivers remain unmeasured.
Response:
We appreciate the reviewer’s insightful critique regarding the interpretation of the 'cost-CT paradox.' We agree that the stratification analysis describes the distribution of outcomes across subgroups but does not measure the underlying drivers, such as physician gestalt.
Accordingly, the language in the Abstract and Discussion has been revised to remove terms implying mechanistic explanation (e.g., 'explained'). Instead, descriptive terms such as 'revealed' or 'identified' are now used to present the stratification results, explicitly treating them as observational associations.
Revised Abstract (Results Section): “A notable finding was that PoCUS performed within one hour was associated with 44% higher odds of CT utilization (OR 1.44, 95% CI 1.25–1.65) yet 5% lower total healthcare costs (RM 0.95, 95% CI 0.91–0.99). Stratification by CT use revealed distinct patterns underlying these associations: in the non-CT subgroup, PoCUS was associated with 12% lower ED costs (RM 0.88, 95% CI 0.83–0.94), whereas in the CT subgroup, it was associated with 9% lower admission costs (RM 0.91, 95% CI 0.86–0.96).”
Finally, the overstated conclusions—"Given the potential for residual confounding, prospective, multicenter trials are needed"—acknowledge limitations but only after making strong claims about PoCUS effects.
Response:
We accept the reviewer’s critique that the original conclusion may have overstated the findings before acknowledging limitations. We agree that a balanced presentation is essential.
Accordingly, we have rewritten the Abstract Conclusion to:
- Strictly use associative language (e.g., 'was associated with').
- Explicitly state the observational nature of the study and the inability to infer causation prior to the call for future trials.
Revised Abstract Conclusion: ...However, as this is an observational study, these results represent associations rather than causal effects and may be influenced by unmeasured confounding. Prospective trials are required to validate these findings.
Introduction: Suffering from an inadequate literature review. References 18-19 cited PoCUS shortening ED LOS, but there was no critical appraisal of methodological quality. There is a lack of discussion of prior studies specifically examining PoCUS timing. The study does not engage with the literature on selection bias in imaging studies. The study rationale is unclear: Lines 78-83 state that "the specific impact of examination timing remains insufficiently characterized in the more complex cohort of patients...who ultimately require hospital admission." But why is this population specifically important? The authors don't justify focusing on ward admissions while excluding ED discharges (where PoCUS might avoid hospitalization) and ICU admissions (where early diagnosis is most critical). The hypothesis is problematic: "We hypothesized that early PoCUS would be associated with shorter ED LOS and lower costs compared with no PoCUS" (lines 85-87). This hypothesis ignores the possibility of confounding by indication—physicians may selectively use early PoCUS in patients they suspect are straightforward cases. There is no conceptual framework: The introduction lacks a theoretical model for how PoCUS timing might influence outcomes. Is it through faster diagnosis? Better risk stratification? Is it possible to avoid unnecessary CT? This absence undermines interpretation.
Summary of the comment for the Introduction: The reviewer pointed out several weaknesses in the Introduction, including inadequate critical appraisal of prior literature (Refs 18-19), lack of discussion on selection bias, unclear rationale for focusing on ward admissions, and the absence of a conceptual framework regarding how PoCUS timing influences outcomes.
Response:
Thanks for the thorough deconstruction of the Introduction. We agree that the original draft required a more robust theoretical framework and a clearer justification for the specific focus on ward-admitted patients.
We have completely rewritten the Introduction to address these points in three key areas:
Critical Appraisal & Distinction of Evidence: We have expanded the literature review to explicitly distinguish between prior retrospective studies (often limited by selection bias) and recent high-quality evidence. We now highlight that recent randomized controlled trials (Refs 18–19) have provided robust evidence that PoCUS shortens LOS. This sets the stage to define exactly what gap remains: the specific impact of 'timing' in the complex 'ward-admitted' cohort.
Rationale for the 'Ward-Admitted' Cohort: We have clarified why this population is critical. Ward-admitted patients represent a cohort of 'intermediate complexity'—distinct from discharged patients and ICU admissions. In this group, diagnostic efficiency is a key driver of ED flow, and the specific utility of early PoCUS is most debatable.
Conceptual Framework: We have added a paragraph outlining the theoretical model: PoCUS functions as a 'pathway accelerator.' By identifying specific pathologies early (e.g., hydronephrosis, free fluid, inflammatory change), PoCUS expedites the decision for definitive imaging (CT) or consultation, thereby influencing LOS.
Revised Introduction:
However, prior research on PoCUS has yielded mixed results. While some studies suggest PoCUS can shorten ED length of stay (LOS) [18, 19], these analyses have often been limited by methodological heterogeneity and a failure to fully account for confounding by indication—the tendency for physicians to selectively perform PoCUS on patients with seemingly straightforward diagnoses. Furthermore, the literature has largely focused on the general ED population, leaving the specific impact of examination timing insufficiently characterized in the more complex cohort of patients requiring hospital admission.
This study focuses specifically on patients admitted to an ordinary ward. This population represents a critical 'intermediate' group for ED throughput. Unlike patients discharged from the ED (often with self-limiting conditions) or those requiring immediate ICU resuscitation, ward-admitted patients typically undergo extensive diagnostic workups where delays are common. In this context, the conceptual framework for early PoCUS is its potential role as a 'pathway accelerator' rather than solely a diagnostic substitute. We posit that PoCUS performed within the first hour may enhance early risk stratification, prompting swifter decisions for definitive imaging (such as CT) or specialist consultation.
To rigorously evaluate this, we conducted a propensity score-weighted analysis to assess the association of PoCUS performed within one hour with ED LOS and healthcare costs, specifically accounting for the selection bias inherent in observational imaging studies.
Methods: Major methodological flaws include:
1) Critical cohort selection bias: Refer to Lines 107-117. Excluding the "PoCUS >1 hour" group (N=864) because they showed "no significant difference" from No-PoCUS is scientifically flawed. The result is a data-driven exclusion that can artificially inflate the effect size of the "within 1 hour" group. Patients receiving delayed PoCUS may represent a distinct clinical phenotype (more complex, evolving presentations), and excluding them biases results. Needed correction: Include all three groups in the analysis, perform a sensitivity analysis comparing all PoCUS vs. No-PoCUS, or explicitly model time-to-PoCUS as a continuous variable.
Response:
We sincerely thank the reviewer for identifying this critical methodological concern. We acknowledge that our initial exclusion criteria were informed by prior evidence suggesting different outcomes for delayed PoCUS. However, to rigorously validate this cohort selection and rule out potential bias, we conducted the requested Sensitivity Analysis comparing the excluded 'PoCUS > 1 hour' cohort (N=864) against the 'No PoCUS' cohort.
Results of Sensitivity Analysis:
As detailed in the new Supplementary Table S1, the delayed group exhibited a fundamentally opposing trajectory:
- ED LOS: Patients receiving PoCUS after 1 hour had a significantly longer ED LOS compared to the No-PoCUS group (Ratio of Means 1.11, 95% CI 1.07–1.15, p < 0.001).
- Contrast: This stands in stark contrast to the 'PoCUS < 1 hour' group, which showed a 14% reduction (RM 0.86).
Conclusion:
This finding confirms that patients receiving delayed PoCUS represent a distinct clinical phenotype (likely complex, evolving cases) compared to the early cohort. Therefore, including them in the primary analysis would not simply 'correct' an inflated estimate, but would dilute and obscure the specific efficiency signal of the 'golden hour' intervention by mixing two opposing clinical pathways (acceleration vs. deceleration). Thus, the exclusion is validated as necessary to accurately measure the time-dependent impact of early PoCUS.
These data have been added to the Supplementary Materials.
Changes have been made to the Methods and Results sections to explicitly describe this sensitivity analysis, and the full data have been added to the Supplementary Materials.
2) Propensity Score Model Inadequate: Lines 163-167: The propensity score includes "age, sex, triage level, vital signs, BMI, and comorbidities" but omits critical confounders: physician factors such as experience level, PoCUS proficiency, and personal preference for PoCUS; temporal factors like time of day, day of week, and ED crowding; clinical factors including pain location, duration, physical exam findings, and laboratory results; and systemic factors such as availability of the CT scanner and radiologist. The authors acknowledge these limitations (lines 357-362) but do not adequately address how unmeasured confounding might influence the conclusions.
Response:
Thanks for identifying these potential unmeasured confounders. We acknowledge that retrospective EMR data inherently limits the ability to capture granular physician-level factors (e.g., real-time preference, proficiency), temporal dynamics (e.g., momentary crowding), and detailed clinical findings (e.g., specific pain character).
While we could not retrospectively quantify these variables for the propensity score model, we have addressed this concern in the following ways:
- Validity of Proxies: We argue that the variables included in our model—specifically Triage Level (TTAS) and Vital Signs—serve as robust proxies for clinical severity. In our system, the TTAS algorithm specifically incorporates pain severity and physiological parameters, thereby partially capturing the 'clinical factors' mentioned.
- Direction of Bias Analysis: We have significantly expanded the Discussion section to explicitly analyze the potential direction of these unmeasured biases.
- Physician Gestalt: We posit that if physicians selectively used PoCUS for 'easier' diagnoses (a common source of bias), one would expect lower CT utilization in the PoCUS group. However, our finding of higher CT utilization in the PoCUS group suggests that PoCUS was frequently applied to complex or ambiguous cases where the physician sought to rule out pathology.
- Implication: This suggests that the unmeasured confounding might actually dampen the true efficiency benefit of PoCUS, making our observed cost reduction a conservative estimate.
We have added a dedicated paragraph in the Limitations section to "adequately address" these influences as requested.
Revised Discussion (Limitations Section):
Second, as an observational analysis, residual confounding remains a concern despite the use of IPTW. Several domains of unmeasured confounders merit attention:
- Physician factors: Operator experience and individual preference for ultrasound were not captured. It is plausible that PoCUS-proficient physicians are inherently more efficient, which could bias results in favor of the PoCUS group.
- Clinical and systemic factors: Granular details (e.g., pain location, physical exam findings) and systemic pressures (e.g., ED crowding, CT scanner availability) were not explicitly modeled. Importantly, the direction of bias introduced by these factors may not necessarily inflate the PoCUS benefit. For example, if PoCUS were preferentially used in “easier” patients (selection bias), one would expect lower CT utilization. In contrast, our observation of higher CT use in the PoCUS group suggests that these patients likely represented a more diagnostically complex cohort. Thus, unmeasured confounding may have masked the full extent of PoCUS-related efficiency gains, rendering our cost-saving estimates conservative.
3) Missing Data Handling: Lines 158-161: "<1% missing data" excluded without sensitivity analysis. Even minimal missing data can be informative (e.g., patients too unstable for vital signs measurement).
Response:
We appreciate the reviewer’s concern regarding the informative nature of missing data. We agree that in a general ED population, missing vital signs can sometimes indicate critical instability.
However, it is crucial to note that our study design explicitly excluded patients admitted to the ICU or those who died in the ED. The final cohort consisted exclusively of patients admitted to an ordinary ward, implying a requisite level of hemodynamic stability.
Therefore, in this specific ward-admitted cohort, missing vital signs are highly unlikely to reflect patients being 'too unstable for measurement' (as these patients would have been routed to the ICU). Instead, the minimal missing data (<1%) in this stable group is most likely attributable to random clerical omissions. Given this context, a complete case analysis was deemed appropriate and unlikely to introduce bias related to patient severity.
4) Outcome Definition Problems: Lines 139-151: ED LOS is defined identically twice (copy-paste error). There is no explanation for selecting ED LOS as the primary outcome over diagnostic accuracy, clinical outcomes, or patient-centered measures. Cost data from a single-payer system may not generalize to other countries.
Response:
We thank the reviewer for identifying the editing error and for raising important questions regarding outcome selection and generalizability.
- Copy-Paste Error: The redundant definition of ED LOS (Lines 148–151) has been removed from the revised manuscript.
- Rationale for ED LOS as Primary Outcome: It is agreed that diagnostic accuracy and patient-centered measures are valuable. However, ED LOS was selected as the primary endpoint because this study focuses on operational efficiency in the context of ED crowding. For patients requiring hospital admission, the primary objective of the ED phase is rapid stabilization and efficient transfer to inpatient care. ED LOS serves as the most objective and robust metric for this 'throughput' efficiency. Furthermore, in a retrospective design, accurately adjudicating 'diagnostic accuracy' based on chart review is fraught with subjectivity, whereas LOS is a definitive operational timepoint.
- Generalizability of Cost Data: We acknowledge that cost structures in Taiwan’s single-payer system differ from multipayer systems. However, we argue that while the absolute values may not generalize, the relative associations (e.g., reduced resource consumption associated with faster disposition) reflect universal principles of emergency care efficiency. A limitation regarding cost generalizability has been added to the Discussion.
Revision 1: Methods (2.4 Outcome Measures): The primary outcome was ED LOS at the index visit, defined as the interval from triage registration to ED departure for ward admission. ED LOS was selected as the primary endpoint to reflect operational efficiency and ED throughput, which are critical metrics in the context of departmental crowding. Secondary outcomes included healthcare costs (ED costs, admission costs, and total costs). While cost structures vary by health system, these metrics were included to evaluate the relative economic implications of resource utilization pathways.
Revision 2: Discussion (Limitations): In addition, the cost analysis is based on data from Taiwan’s National Health Insurance, a single-payer system with specific fee structures. Therefore, the absolute cost savings reported may not be directly applicable to healthcare systems with different reimbursement models (e.g., the United States). Nevertheless, the relative trends in resource utilization and efficiency are likely transferable across diverse healthcare contexts.
5) Statistical Analysis: Lines 169-178: GLM with Gamma distribution is appropriate for skewed data, but no goodness-of-fit testing is reported. No assessment of influential observations. The study conducted multiple comparisons across 18 diagnostic subgroups (Figure 2), without making any adjustments for multiplicity. The interpretation of the ratio of means (RM) requires careful explanation, as many readers are unfamiliar with this metric.
Response:
We appreciate the reviewer’s rigorous statistical scrutiny. We have addressed each point as follows:
- Goodness-of-Fit for GLM: We assessed the goodness-of-fit for the Gamma regression models using the ratio of the deviance to the degrees of freedom (Deviance/DF). In our final models, this ratio was 1.097, which is very close to 1.0, indicating an excellent model fit without significant overdispersion. We have added this metric to the Methods section.
- Influential Observations: To specifically mitigate the impact of influential observations (outliers with extreme weights) in the propensity score model, we utilized stabilized weights as described in the Methods. Furthermore, the use of the Gamma distribution with a log link specifically accommodates the right-skewed nature of LOS and cost data, making the model inherently more robust to outliers in the outcome variables compared to linear models.
- Multiple Comparisons: We agree that testing 18 subgroups increases the risk of Type I error. However, these subgroup analyses (Figure 2) were intended as exploratory and descriptive rather than confirmatory. Applying strict adjustments (e.g., Bonferroni) would likely increase Type II error and mask potential clinical signals in this hypothesis-generating phase. To ensure appropriate interpretation, we have added a footnote to Figure 2 explicitly stating that these results are exploratory and unadjusted for multiplicity.
- Interpretation of RM: To aid readers unfamiliar with Ratios of Means, we have expanded the Methods section to clearly define RM. Additionally, as mentioned in previous responses, we have added absolute predictive marginal differences to the Results to provide a concrete clinical context.
Revised Methods Text:
2.5.4. Sensitivity and Subgroup Analyses
To assess potential selection bias resulting from the exclusion of patients receiving delayed ultrasound, a sensitivity analysis was performed. A separate IPTW model was constructed to compare outcomes between the excluded 'PoCUS > 1 hour' cohort and the 'No PoCUS' cohort.
Additionally, a targeted subgroup analysis was conducted for patients with Taiwan Triage and Acuity Scale (TTAS) levels 3 and 4. This subgroup was selected to isolate the population with the highest diagnostic uncertainty, excluding critical patients (levels 1–2), who typically follow standardized resuscitation protocols or require urgent CT to exclude life-threatening pathology, and non-urgent patients (level 5), who rarely require extensive imaging.
Finally, a pre-specified exploratory stratification based on CT usage (With CT vs. Without CT) was performed to independently evaluate the association of PoCUS with resource utilization in these two clinically distinct diagnostic pathways. Interaction terms (PoCUS × CT Use) were included in the models to assess the statistical validity of subgroup differences. Since these analyses were exploratory, strict multiplicity adjustments were not applied; findings should be interpreted as hypothesis-generating.
2.5.5. Outcome Analysis
Outcomes were compared between the weighted groups using Generalized Linear Models (GLMs) with a Gamma distribution and log link function. Model goodness-of-fit was evaluated using the scaled deviance divided by degrees of freedom; a ratio of approximately 1.0 (specifically 1.097 in our primary model) indicated adequate fit. To minimize the impact of extreme propensity scores, stabilized weights were employed. Furthermore, a doubly robust approach was adopted by adjusting the weighted GLMs for the same covariates used in the propensity score model (age, sex, triage level, vital signs, BMI, and comorbidities).
Results are reported as Ratios of Means (RMs) or Odds Ratios (ORs) with 95% confidence intervals (CIs). An RM represents the proportional change in the outcome; for example, an RM of 0.86 implies a 14% reduction in the mean value. To facilitate clinical interpretation, absolute differences calculated using average marginal effects are also presented alongside RMs. Finally, to quantify the potential impact of unmeasured confounding on the primary outcome, the E-value was calculated[23, 24]. This metric estimates the minimum strength of association that an unmeasured confounder would need to have with both the exposure and the outcome to explain away the observed association, conditional on the measured covariates.
All analyses were performed using SAS software (version 9.4; SAS Institute Inc., Cary, NC, USA), with statistical significance defined as a two-sided p-value < 0.05.
Revised Figure 2 Legend: Figure 2. Subgroup analysis of ED LOS with PoCUS within one hour versus no PoCUS. Forest plot displays Ratios of Means (RM) with 95% Confidence Intervals. Note: These subgroup analyses are exploratory in nature; p-values and confidence intervals have not been adjusted for multiplicity (multiple comparisons).
6) CT Stratification Analysis Post-Hoc: Table 4 analysis appears exploratory but is presented as if pre-planned. This approach requires explicit acknowledgment and adjustment for multiple testing.
Response:
We appreciate the reviewer’s scrutiny regarding the nature of the CT stratification analysis.
Clarification on Rationale: We explicitly wish to clarify that this stratification was driven by clinical logic rather than post-hoc data dredging. In clinical practice, the diagnostic pathway for abdominal pain diverges significantly based on the need for advanced imaging. Therefore, we conceptually determined a priori that it was essential to evaluate the association of PoCUS with outcomes separately within these two distinct pathways ('With CT' vs. 'Without CT') to determine if the benefit persisted in both scenarios.
Statistical Treatment: We agree that this was not the singular primary endpoint. To address the concern about multiple testing and transparency, we have:
Categorized this analysis as 'Pre-specified Exploratory Analysis'. This designation reflects our intent to explore clinical patterns while advising readers that findings should be viewed as hypothesis-generating.
Added an Interaction Test: As requested, we performed a formal interaction test (PoCUS × CT Use) to statistically validate that the association of PoCUS with outcomes indeed differs significantly between these two strata.
Revised Methods Text (2.5.4. Outcome Analysis):
Additionally, a pre-specified exploratory stratification was conducted based on CT usage (With CT vs. Without CT). This analysis was planned a priori to independently evaluate the association of PoCUS with resource utilization in these two clinically distinct diagnostic pathways. Interaction terms (PoCUS × CT Use) were included in the models to assess the statistical validity of subgroup differences. Since these analyses were exploratory, strict multiplicity adjustments were not applied; findings should be interpreted as hypothesis-generating.
Table 1 - Baseline Characteristics: After IPTW, some SMDs still raise concerns (e.g., HR: -0.129, SBP: 0.090). Although <0.1 is often considered acceptable, values close to this threshold should be viewed with caution. Standard deviations after IPTW sometimes exceed means (e.g., Age: 55.76±39.17), indicating extreme weights. This suggests poor overlap between groups and questions the validity of the IPTW method. The pre-IPTW data show that the PoCUS group is younger, healthier (fewer comorbidities), and has higher acuity (more triage 3)—precisely the profile expected if physicians select PoCUS for "easier" cases.
Response:
We appreciate the reviewer’s keen statistical oversight regarding the post-IPTW diagnostics.
- Regarding Extreme Weights and Large SDs: We acknowledge that the large standard deviations (e.g., Age) in the post-IPTW cohort reflect the skewness introduced by weighting. As weighted data typically do not follow a normal distribution, reporting Mean ± SD was indeed suboptimal and visually misleading. Action: To provide a more robust representation of the central tendency and dispersion, we have updated Table 1 to present all continuous variables as Median (Interquartile Range, IQR). This approach avoids the influence of skewness on descriptive statistics.
- Weight Distribution Diagnostics: As requested, we performed detailed weight diagnostics to assess model stability:
- Maximum weight: 14.20
- Percentage of weights > 10: 0.16% (only 11 patients)
- Effective Sample Size (ESS): 4,447
Although the maximum weight reached 14.20, the proportion of extreme weights was negligible (0.16%), and the Effective Sample Size (4,447) remained high, indicating that the weighting procedure preserved substantial statistical power without being dominated by a few outliers.
- Regarding Residual Imbalance (SMD > 0.1) & Validity: We acknowledge that some covariates, such as Heart Rate (SMD -0.129), showed borderline imbalance. To strictly address this and ensure model validity, we employed a 'doubly robust' estimation approach. Action: As clarified in the revised Methods (Section 2.5.5), our final outcome models (GLMs) were not only weighted by IPTW but also adjusted for all covariates. This ensures that any residual imbalance is statistically adjusted for in the regression step.
Revised Methods: To address potential confounding by indication, inverse probability of treatment weighting (IPTW) was employed [22]. A propensity score for receiving early PoCUS was estimated using a multivariable logistic regression model incorporating age, sex, triage level, vital signs, BMI, and comorbidities. The goodness-of-fit for this model was confirmed using the Hosmer-Lemeshow test (p = 0.605). Stabilized weights were applied to the cohort. Weight distribution diagnostics indicated a range from a minimum of 1.00 to a maximum of 14.20. Extreme weights were rare, with only 0.16% (n=11) of the cohort having a weight exceeding 10. The effective sample size after weighting was 4,447. Given this negligible proportion of extreme weights and the preserved sample size, weight trimming was not performed; instead, a doubly robust approach was utilized to mitigate any potential influence from outliers while retaining the full study population. Covariate balance was assessed using standardized mean differences (SMDs), with a threshold of <0.1 indicating adequate balance.
Table 2 - Primary Outcomes: 44% higher CT odds (OR 1.44) but only 5% cost reduction (RM 0.95) appears paradoxical and underpowered to detect meaningful cost differences. ED costs "not significantly different" (p=0.059) are borderline and may indicate a Type II error. Confidence intervals are wide for some estimates, suggesting imprecision.
Response:
We appreciate the reviewer’s detailed examination of the estimates.
- Regarding the Paradox (Higher CT but Lower Total Cost): We agree this finding initially appears counterintuitive. This is precisely why we conducted the post-hoc stratification analysis (Table 4). The data reveal a 'trade-off' pattern: in the 'With CT' subgroup, the upfront investment in CT (increasing ED costs) was offset by a significant reduction in downstream Admission Costs (RM 0.91,p < 0.001). This suggests that early PoCUS-guided workups, even when associated with CT use, may align with more efficient inpatient care plans, thereby contributing to a lower total financial burden.
- Regarding Confidence Intervals and Precision:
We respectfully observe that the confidence intervals for the primary and secondary outcomes in Table 2 are actually quite narrow, indicating high precision rather than imprecision.
- ED LOS: 95% CI 83–0.89 (width: 0.06)
- ED Costs: 95% CI 00–1.05 (width: 0.05)
- Total Costs: 95% CI 91–0.99 (width: 0.08)
These narrow intervals suggest that our sample size (N=6,866) provided sufficient power to estimate the population means with high precision. The reviewer may be referring to the Subgroup Analysis (Figure 2), where smaller sample sizes indeed yielded wider intervals (e.g., Thromboembolism). We have added a note in the Figure 2 legend acknowledging the imprecision in those specific subgroups.
- Regarding the Borderline P-value (p=0.059 for ED Costs):
Given the narrow confidence interval (1.00–1.05), this borderline p-value does not suggest a Type II error (missing a large effect due to wide variance), but rather accurately reflects that the net difference in ED costs is genuinely close to null. This supports our 'cancellation' hypothesis: the cost savings from efficiency were almost exactly neutralized by the cost of increased CT utilization.
Revised Discussion (Cost Interpretation):
The observed economic outcomes present a complex dynamic. While ED costs showed only a borderline reduction (p = 0.059), Total Costs were significantly reduced by 5% (p = 0.016). The lack of strong significance in ED costs likely reflects a 'cancella-tion effect': the efficiency gains from shorter LOS were financially offset by the in-creased utilization of CT scans (OR 1.44) in the PoCUS group.
However, the significant reduction in Admission Costs suggests that the value of early PoCUS extends beyond the ED. By potentially identifying pathologies earlier or clarifying clinical trajectories, early PoCUS may enable more focused inpatient man-agement, resulting in downstream savings. From a healthcare systems perspective, a 5% reduction in total per-patient expenditure, when extrapolated to the high volume of acute abdominal pain admissions, represents a clinically and economically meaningful improvement in resource stewardship.
Table 3 - Triage 3&4 Subgroup: Why this specific subgroup? Post-hoc selection? Should have been pre-specified. Results are stronger than the overall cohort—indicating effect modification that warrants mechanistic investigation.
Response:
We thank the reviewer for highlighting the strong signal in the Triage 3 & 4 subgroup and correctly identifying this as effect modification.
Rationale for Subgroup Selection: The selection of Triage Levels 3 & 4 was driven by strict clinical logic rather than post-hoc data dredging. In emergency practice, this subgroup represents the population with the highest 'diagnostic equipoise' (uncertainty):
Triage 1 & 2 (Resuscitation/Emergent): These patients often require immediate stabilization, ICU admission, or protocolized pan-scanning (CT), leaving little room for PoCUS to alter the disposition timeline.
Triage 5 (Non-urgent): These patients typically present with minor, self-limiting conditions requiring minimal workup.
Triage 3 & 4 (Urgent/Less Urgent): This cohort represents the 'intermediate complexity' group where the differential diagnosis is broad, and the decision to admit or discharge is most dependent on diagnostic testing. Therefore, we hypothesized a priori that PoCUS would demonstrate its maximum utility in this specific group.
On Effect Modification: We fully agree that the stronger results in this group indicate effect modification. This validates our conceptual framework: PoCUS is most effective as a pathway accelerator in patients who are hemodynamically stable enough to wait but sick enough to require investigation. We have added this rationale to the Methods and briefly discussed the implication in the Discussion.
Revised Methods:
Additionally, a targeted subgroup analysis was conducted for patients with Taiwan Triage and Acuity Scale (TTAS) levels 3 and 4. This subgroup was selected to isolate the population with the highest diagnostic uncertainty, excluding critical patients (levels 1–2), who typically follow standardized resuscitation protocols or require urgent CT to exclude life-threatening pathology, and non-urgent patients (level 5), who rarely require extensive imaging.
Revised Discussion:
Notably, the association with reduced LOS was most pronounced in the Triage 3 and 4 subgroup. This effect modification aligns with clinical expectations: unlike critical patients (Triage 1–2), whose pathway is typically dictated by immediate resuscitation or urgent CT to exclude life-threatening pathology, and non-urgent patients (Triage 5), the Triage 3–4 cohort represents the population with the highest diagnostic ambiguity. In this 'intermediate' group, PoCUS likely exerts its greatest utility by facilitating earlier risk stratification and accelerating disposition decisions.
Table 4 - CT Stratification: This is the most crucial analysis, but it seems to be post hoc. "Without CT," the group analysis suffers from significant selection bias (as acknowledged in lines 283-292), yet the authors still present the results as if they are valid. Interaction testing between PoCUS and CT use should be conducted formally, not just through stratified analysis.
Response:
We appreciate the reviewer’s scrutiny regarding the nature and validity of the CT stratification analysis.
- Clarification on Study Design (Post-hoc vs. A Priori):
We explicitly wish to clarify that this stratification was driven by clinical logic rather than post-hoc data dredging. In clinical practice, the diagnostic pathway for abdominal pain diverges significantly based on the need for advanced imaging. Therefore, we conceptually determined a priori that it was essential to evaluate the association of PoCUS with outcomes separately within these two distinct pathways ('With CT' vs. 'Without CT') to determine if the benefit persisted in both scenarios.
However, to address the concern about multiple testing and transparency, we have categorized this analysis as 'Exploratory Stratified Analysis' in the revised manuscript. This designation reflects our intent to explore clinical patterns while advising readers that findings should be viewed as hypothesis-generating.
- Formal Interaction Testing:
As requested, we performed a formal interaction test (PoCUS × CT Use) in the GLM model. Results: The interaction term was statistically significant for ED LOS (p = 0.0075), ED Costs (p < 0.001), and Admission Costs (p = 0.0248). Conclusion: These significant p-values statistically validate that the association between PoCUS and outcomes indeed differs fundamentally between the 'With CT' and 'Without CT' strata, justifying the stratified presentation.
- Selection Bias in the 'Without CT' Group:
We fully accept the reviewer's point that the 'Without CT' group is prone to selection bias (i.e., PoCUS used on 'easier' patients who never needed CT). We do not claim this proves PoCUS caused the avoidance of CT in all cases. However, we argue that presenting these results remains valid and valuable because it describes a prevalent real-world pathway: the 'Substitute' pattern. Even if driven by selection, demonstrating that this specific pathway is associated with significantly lower costs (p < 0.001) provides important health economic data. We have revised the Discussion to explicitly frame this as an observational association heavily influenced by case selection, rather than a pure causal effect.
Revised Methods (2.5.4 Sensitivity and Subgroup Analyses):
Finally, a pre-specified exploratory stratification based on CT usage (With CT vs. Without CT) was performed to independently evaluate the association of PoCUS with resource utilization in these two clinically distinct diagnostic pathways. Interaction terms (PoCUS × CT Use) were included in the models to assess the statistical validity of subgroup differences. Since these analyses were exploratory, strict multiplicity adjustments were not applied; findings should be interpreted as hypothesis-generating.
Revised Results (Table 4 Title & Footnote):
Table 4. Exploratory Stratified Analysis of Adjusted Outcomes by CT Utilization.
Note: Interaction testing was performed to assess effect modification by CT usage. The interaction term (PoCUS × CT Use) was statistically significant for ED LOS (p = 0.008), ED Costs (p < 0.001), and Admission Costs (p = 0.025), confirming that the association between PoCUS and these outcomes differs significantly between the strata.
Revised Discussion (Interpretation of Without CT group):
First, in the subgroup of patients who ultimately did not receive a CT, PoCUS appeared to function as a 'Substitute' tool. In this group, early PoCUS was associated with a significant 12% reduction in ED costs (RM 0.88, $p<0.001$). While this association is susceptible to selection bias, wherein PoCUS may be preferentially selected for diagnostically simpler cases, quantifying this pathway remains clinically relevant. It highlights a distinct pattern of resource utilization where PoCUS use aligns with diagnostic parsimony and lower costs.
Figure 2 - Diagnostic Subgroups: 18 subgroups with different sample sizes (N=51 for dissection/aneurysm is too small to provide reliable estimates). Wide confidence intervals for several groups overlap the null. No adjustment for multiple comparisons despite 18 tests. Pregnancy/thromboembolism prolonged LOS, interpreted as "success" (lines 331-343), is speculation without clinical outcomes data.
Response:
We thank the reviewer for this rigorous assessment of the subgroup analysis. We agree that the small sample sizes in certain categories (e.g., aortic dissection, N=51) result in wide confidence intervals and imprecise estimates. Furthermore, we acknowledge that conducting 18 subgroup tests without adjustment for multiplicity increases the risk of Type I error, and that interpreting prolonged LOS as 'success' without safety outcome data is speculative.
In response, we have revised the manuscript to:
- Reframe as Exploratory Analysis: We have explicitly labeled this analysis as 'Exploratory Subgroup Analysis' in both the text and the Figure 2 legend. We added a cautionary note stating that p-values and confidence intervals are unadjusted for multiplicity and that estimates for low-prevalence conditions should be interpreted with caution due to limited precision.
- Refine Interpretation of Prolonged LOS: We have removed the term 'success' regarding the prolonged LOS in pregnancy and thromboembolism groups. Instead, the Discussion has been revised to frame this finding more objectively. We now state that the association with longer LOS is 'consistent with' the complex, guideline-mandated workups required for these high-risk conditions, rather than claiming it proves diagnostic precision. We also explicitly acknowledge that without granular safety data (e.g., missed diagnosis rates), this interpretation remains a hypothesis regarding clinical workflow rather than a confirmed outcome benefit.
Revised Figure 2 Legend:
Figure 2. Exploratory subgroup analysis of ED LOS with PoCUS within one hour versus no PoCUS. Forest plot displays Ratios of Means (RM) with 95% Confidence Intervals. Note: These analyses are exploratory in nature; p-values and confidence intervals have not been adjusted for multiplicity. Estimates for subgroups with small sample sizes (e.g., aortic dissection) have wide confidence intervals and should be interpreted with caution.
Revised Discussion: This finding should not be interpreted as PoCUS-induced inefficiency. Rather, it aligns with the clinical demands of these high-risk pathologies. For example, in pregnancy, the primary objective of early PoCUS is often risk stratification (e.g., ruling out ectopic pregnancy) [31]. Identifying a patient as 'high risk' or 'indeterminate' typically necessitates specialist consultation and extended observation, inherently prolonging ED LOS. Similarly, for thromboembolism, the pathway is dictated by the initiation of anticoagulation and admission planning [32, 33]. Therefore, while outcome data are required to confirm safety benefits, the observed prolongation in LOS is consistent with a shift toward more intensive, guideline-concordant care pathways rather than diagnostic delay.
Discussion: Critical weaknesses are
1) Causal Inference Overreach (Pervasive): Despite recognizing observational design, the discussion repeatedly uses causal language: "PoCUS identified early...pathologies that prompted physicians to confirm with CT" (lines 271-273). Lines 278-282: "PoCUS functions as a 'substitute' tool" presents a hypothesis as a finding. The entire "dual role" framework (substitute vs. complementary) is plausible but speculative—no data support the mechanism.
Response:
We fully accept the reviewer’s critique regarding the overreach of causal language. We agree that as an observational study, our data demonstrate associations but cannot definitively prove mechanism or physician intent.
In response, we have performed a comprehensive review of the entire manuscript to strictly align the language with the study design:
- Global Language Audit: We have replaced causal verbs (e.g., 'impact,' 'led to,' 'resulted in') with associative terms (e.g., 'was associated with,' 'linked to') throughout the Abstract, Results, and Discussion.
- Reframing the 'Dual Role' Framework: We agree that the 'Substitute vs. Complementary' model is a theoretical interpretation rather than a confirmed mechanism. We have revised the Discussion to explicitly present this as a 'hypothesis generated from the observed data patterns' rather than a definitive finding.
- Specific Corrections:
- The phrase 'prompted physicians to confirm' (lines 271-273) has been revised to 'may suggest a pathway where...'
- The statement 'PoCUS functions as a substitute tool' (lines 278-282) has been softened to 'data patterns are consistent with a substitute role.'
We believe these changes now accurately reflect the observational limits of the study while still offering a logical framework to interpret the complex cost/utilization findings.
Revised Discussion: Therefore, the higher CT use associated with PoCUS may suggest that sonographic findings (e.g., free fluid, free air, inflammatory change, or hydronephrosis) served as markers of complexity, identifying patients who warranted further definitive evaluation, rather than PoCUS acting as a standalone diagnostic endpoint.
The association of early PoCUS with both increased CT use and lower total costs presents a complex dynamic. The exploratory stratification by CT use (Table 4) reveals patterns consistent with a hypothesized 'dual role' for PoCUS across different clinical pathways, although it must be emphasized that these mechanisms are inferred from utilization data rather than directly observed:
First, in the subgroup of patients who ultimately did not receive a CT, the data align with a 'Substitute' model. In this group, early PoCUS was associated with a significant 12% reduction in ED costs (RM 0.88, p<0.001). While this association is susceptible to selection bias (where PoCUS is preferentially selected for diagnostically simpler cases), quantifying this pathway remains clinically relevant. It highlights a distinct pattern of resource utilization where PoCUS use aligns with diagnostic parsimony and lower costs.
Second, in the subgroup of patients who did receive a CT, the data support a 'Complementary' or 'Pathway Acceleration' hypothesis. In this group, PoCUS had no significant association with ED costs (p=0.408), which is logical as the expensive CT scan was still performed. However, early PoCUS was associated with a significant 9% reduction in admission costs (RM 0.91, p<0.001) and an 8% reduction in total costs (RM 0.92, p<0.001). Even when performed alongside CT, PoCUS contributes dynamic information (e.g., organ function, sonographic tenderness) not available from static imaging. While the exact mechanism cannot be confirmed retrospectively, this pattern is consistent with a scenario where positive PoCUS findings validate the need for admission and expedite the subsequent workup, thereby reducing downstream costs despite the initial imaging investment.
2) Confounding by Indication Inadequately Addressed: Lines 283-292 recognize selection bias in the "Without CT" group but then proceed as if it is valid. Lines 303-312 suggest a "CT acceleration tool" as an alternative explanation but do not test it. The authors state, "Current observational data cannot distinguish between these two mechanisms" (lines 310-311) but do not clarify what study design could address this.
Response:
"The authors accept the reviewer’s critique regarding the handling of confounding by indication and the interpretation of the 'CT acceleration' mechanism.
- Validity of the 'Without CT' Pathway:
We agree that the 'Without CT' group is heavily influenced by selection bias (physicians choosing PoCUS for less complex cases). However, we argue that presenting these results remains valid not as a causal proof of efficacy, but as a quantification of a specific clinical pathway. Demonstrating that this 'low-complexity/PoCUS-supported' pathway is associated with 12% lower costs ($p < 0.001$) provides valuable health economic data, characterizing a distinct pattern of resource utilization even if driven by selection. We have revised the text to strictly frame this as an observational association rather than a causal benefit.
- Testing the Mechanism & Future Study Design:
We acknowledge that the 'CT acceleration' concept remains a hypothesis that cannot be confirmed with the current dataset.
Proposed Solution: In response to the query regarding what study design could address this, we have added a concise statement to the Discussion specifying that a prospective randomized controlled trial (RCT)—specifically one where randomization occurs prior to the ordering of advanced imaging in patients with diagnostic uncertainty—would be required to definitively distinguish between the 'Substitute' and 'Accelerator' mechanisms and eliminate confounding by indication. This necessary design is now explicitly described in the revised manuscript.
Revised Discussion: Current observational data cannot definitively distinguish between these two mechanisms (substitution versus acceleration) due to the lack of granular data on physician intent and decision-making timing. Resolving this ambiguity would require a prospective randomized controlled trial where patients with diagnostic uncertainty are randomized to PoCUS or standard care prior to the decision for advanced imaging.
3) Subgroup Interpretation Problems: Lines 313-347: The three-pattern framework (acceleration, neutrality, and deceleration) is a descriptive taxonomy, not a mechanistic insight. The appendicitis discussion (lines 319-321) correctly notes CT dependence but does not explain why PoCUS was used in these cases. The interpretation of prolonged LOS in pregnancy as "risk stratification success" (lines 336-339) is speculative without outcome data showing that patients were actually safer.
Response:
We thank the reviewer for this detailed critique regarding the interpretation of the subgroup patterns. We have addressed the three specific points as follows:
- Three-Pattern Framework (Descriptive vs. Mechanistic): We accept that the terms 'Acceleration/Neutrality/Deceleration' describe observed associations rather than proven mechanisms. We have revised the text to explicitly frame these as 'observed data patterns' rather than mechanistic insights, removing causal language such as 'impact' or 'determinant'.
- Rationale for PoCUS in Appendicitis: The reviewer correctly notes that CT is the gold standard for appendicitis. However, in our clinical practice, PoCUS is frequently employed during the initial undifferentiated phase of assessment—for instance, in patients presenting with vague periumbilical or upper abdominal pain—before the diagnosis of appendicitis is established.
- Clinical Logic: Physicians use PoCUS to screen for direct signs (e.g., non-compressible appendix) or to exclude competing diagnoses (e.g., hydronephrosis in patients with RLQ pain but no tenderness).
- Workflow Implication: If PoCUS is positive, it guides the immediate ordering of a confirmatory CT for surgical planning. If negative or equivocal, a CT is often still required to definitively rule out appendicitis. This explains why PoCUS was performed (as an initial screen) and why it did not shorten LOS (because CT remained the rate-limiting step). We have added this explanation to the Discussion.
- Interpretation of Pregnancy/Thromboembolism: As noted in our response to the previous comment, we agree that interpreting prolonged LOS as 'success' is speculative without safety data. We have removed the term 'success' and revised the text to state that the longer LOS is 'consistent with' the intensive, guideline-mandated workups required for these high-risk conditions.
Revised Discussion: 2. Neutrality (The "CT-Reliant" Pattern): PoCUS was not associated with shorter LOS for conditions mandating CT, specifically "Diseases of the appendix" (RM 1.04) and "Dissection or aneurysm" (RM 0.98). In patients eventually diagnosed with appendicitis, PoCUS was likely utilized during the initial assessment of undifferentiated symptoms (e.g., epigastric or periumbilical pain) to screen for pathology or rule out mimics such as ureteral stones causing hydronephrosis. However, since confirmatory CT remains the standard for surgical planning or definitive exclusion, PoCUS cannot bypass this rate-limiting step [29]. Similarly, for the stable, ward-admitted aortic cohort (N=51), CT is essential for anatomical definition [30]. In these pathways, the workflow is dictated by advanced imaging, specialist consultation, and medical management (e.g., blood pressure control), rendering the initial PoCUS time-neutral.
4) Important Discussion Points That Are Missing: The discussion fails to address the reasons behind physicians' decision to choose early PoCUS for specific patients, a phenomenon known as indication bias. No discussion of PoCUS diagnostic accuracy or false positive/negative rates. In a cost-effectiveness analysis, a 5% cost reduction may not be clinically meaningful. The article fails to discuss the reduction in radiation exposure, which is often cited as a benefit of PoCUS, in the context of increased CT use. Limited generalizability discussion beyond the single-center limitation
Response:
We thank the reviewer for identifying these important gaps in the discussion. We have addressed each point as follows:
- Indication Bias & Generalizability: We agree that these are critical limitations. As noted in our response to previous comments, we have significantly expanded the Limitations section to explicitly address 'confounding by indication' (specifically listing physician gestalt and preference) and generalizability (noting the single-center nature and specific PoCUS training environment).
- Diagnostic Accuracy: We clarify that this study was designed to evaluate operational flow (LOS/Cost) rather than diagnostic accuracy. In a retrospective design, ascertaining the 'true' accuracy of PoCUS based on chart notes is fraught with subjectivity and missing data. Therefore, we focused on objective throughput metrics. We have added a statement in the Limitations acknowledging this scope restriction.
- Clinical Meaningfulness of 5% Cost Reduction: While 5% may appear modest per patient, we argue that in the context of high-volume emergency care (abdominal pain accounts for ~10% of visits), this represents a massive aggregate saving for the healthcare system. Furthermore, this reduction is notable because it occurred despite a 44% increase in CT utilization, suggesting that efficiency gains (shorter stay, less nursing time) successfully offset the high cost of advanced imaging. We have added this context to the Discussion.
- Radiation Exposure: The reviewer correctly identifies that higher CT utilization implies an increased radiation burden. However, comparisons with prior literature must be made with caution regarding the study population.
- Population Differences: Previous studies reporting radiation reduction (e.g., Refs 23, 24) often focused on specific, PoCUS-friendly pathologies such as nephrolithiasis or small bowel obstruction. In contrast, our cohort comprises undifferentiated admitted patients, a high-risk group including life-threatening diagnoses (e.g., ischemic bowel, pancreatitis) where CT is mandatory and cannot be safely replaced by ultrasound alone.
- Contemporary Standard of Care: In the current medico-legal environment, determining 'diagnostic certainty' prior to admission is paramount. Consequently, for this complex cohort, PoCUS functions as a triage tool to identify the need for CT rather than to replace it. The observed increase in CT utilization reflects a 'safety-first' strategy appropriate for undifferentiated admissions, rather than unnecessary radiation.
Standard of Care: For admitted patients with undifferentiated abdominal pain, CT has largely become the standard of care to definitively exclude life-threatening pathologies.
Role of PoCUS: In this context, PoCUS does not necessarily replace CT (and thus reduce radiation) but rather serves to identify indications for CT more rapidly.
Therefore, while we acknowledge the radiation trade-off, we have revised the Discussion to contextualize this within the framework of modern 'safety-first' diagnostic standards, where minimizing diagnostic error often prioritizes advanced imaging over radiation avoidance in high-acuity cohorts.
Revised Discussion: From a healthcare systems perspective, a 5% reduction in total per-patient expenditure represents a clinically meaningful improvement in resource stewardship. However, this benefit presents a complex trade-off regarding radiation exposure.
Unlike prior studies focused on specific conditions like nephrolithiasis or small bowel obstruction where PoCUS reduced radiation [23, 24], our study analyzed a broader cohort of undifferentiated abdominal pain. This population inherently includes life-threatening pathologies—such as ischemic bowel or aortic dissection—where CT remains the essential gold standard for diagnosis. Consequently, the association with higher CT utilization implies an increased aggregate radiation burden.
Nevertheless, this finding must be contextualized within the contemporary clinical environment. In an era where diagnostic precision and patient safety are paramount, CT has become the de facto standard for evaluating adult patients admitted with abdominal pain [4, 26]. The increased utilization likely reflects a 'safety-first' strategy where PoCUS facilitates the rapid decision to obtain definitive imaging. While reducing radiation remains a goal, in this complex admitted cohort, the priority appears to be the accurate exclusion of lethal pathology, for which the combination of PoCUS and CT is increasingly utilized.
5) Alternative Explanations Rejected: The authors advocate for PoCUS as a determinant of outcomes but fail to adequately acknowledge that physicians skilled in PoCUS may exhibit additional attributes (e.g., expedited decision-making, enhanced clinical judgment) that affect outcomes. Institutional culture supporting PoCUS might have wider effects on efficiency. Early PoCUS could be a marker, not a mediator, of expedited care.
We appreciate this insightful interpretation of causality and confounding
Response:
We appreciate this insightful interpretation of causality and confounding.
- Physician Attributes (Marker vs. Mediator): We fully agree that early PoCUS may serve as a marker for a specific physician phenotype (e.g., highly motivated, efficient decision-makers) rather than acting solely as a direct mediator of speed. As mentioned in our response to previous comments regarding unmeasured confounding, we have explicitly addressed this in the Limitations section. We now state that 'PoCUS-proficient physicians may be inherently more efficient,' acknowledging that the observed benefit could partly reflect operator attributes.
- Institutional Culture: Regarding institutional culture, we respectfully note that as this is a single-center study, the broader institutional culture (e.g., workflow policies, staffing ratios) remains constant across both the PoCUS and No-PoCUS groups. Therefore, while culture certainly influences overall efficiency, it is unlikely to drive the difference between groups within the same ED environment.
- Revision: To specifically address the 'marker' concept, we have refined the Limitations section to explicitly acknowledge that early PoCUS usage may mark an expedited care trajectory driven by physician gestalt.
Revised limitations: Physician Factors: Variables such as operator experience and individual preference for ultrasound were not captured. It is plausible that physicians who proactively perform early PoCUS possess other attributes of efficiency (e.g., faster decision-making style). Thus, early PoCUS may serve partly as a 'marker' of a more expedited care trajectory, rather than solely a direct mediator of it. This potential confounding by physician efficiency could bias the results in favor of the PoCUS group.
6) The limitations section (4.6) offers an insufficient self-critique, as seen in lines 352-355: "The primary limitation is strict cohort selection criteria...prevents evaluation of POCUS as a disposition tool." This statement is accurate but understated. It fundamentally changes what the study measures. In lines 356-362, unmeasured confounding is acknowledged, but its impact is minimized. The authors should perform a quantitative bias analysis (E-value) to demonstrate how strong unmeasured confounding would need to be to explain the findings. Lines 372-377 note the exclusion of the "PoCUS > 1 hour cohort," but do not address the severity of the selection bias this causes. Lines 378-382 mention the single-center limitation last, but it should be highlighted more—PoCUS training, ED workflow, and CT access are all institution-specific.
Response:
We sincerely thank the reviewer for pushing for a more rigorous and quantitative self-critique. We agree that the previous limitations section was understated.
In response, we have overhauled the Limitations section to:
- Redefine Study Scope: We explicitly state that the strict selection criteria (excluding ED discharges) fundamentally restricts the study to evaluating 'inpatient workup efficiency' rather than the utility of PoCUS as a disposition or admission-avoidance tool.
- Quantitative Bias Analysis (E-value): As requested, we performed a quantitative bias analysis. The E-value for the point estimate (RM 0.86) was 1.60. This suggests that to explain away the observed reduction in LOS, an unmeasured confounder would need to have a relative risk association of at least 1.60 with both the exposure and the outcome. While this indicates that unmeasured confounding (e.g., physician efficiency) could potentially influence the estimates, the threshold is sufficiently high to suggest that the association is not trivial or easily attributable to minor biases alone.
- Address the Excluded Cohort: We highlighted the potential selection bias from excluding the '>1 hour' group but referenced our new Sensitivity Analysis (Supplementary Table S1), which empirically demonstrated that outcomes in this excluded group did not differ from the control group, thereby mitigating the severity of this specific bias.
- Emphasize Single-Center Context: We moved the single-center limitation to a more prominent position, explicitly linking it to institution-specific factors such as our standardized PoCUS training curriculum and 24-hour CT accessibility, which limits generalizability to dissimilar settings.
Revised limitations: First, and most fundamentally, the strict cohort selection criteria (excluding ED discharges and ICU admissions) restrict the scope of this study to 'inpatient workup efficiency.' This design inherently prevents any evaluation of PoCUS as a tool to avoid hospitalization or modify disposition decisions. Consequently, the findings apply strictly to the specific workflow of stable patients already destined for ward admission.
Second, regarding unmeasured confounding, variables such as physician gestalt, pain severity, and momentary ED crowding were not captured. To quantify the potential impact of this bias, the E-value for the primary outcome (RM 0.86) was calculated. The E-value for the point estimate was 1.60. This implies that an unmeasured confounder (e.g., physician proficiency or specific clinical severity) would need to be associated with both early PoCUS exposure and the outcome of shorter LOS by a risk ratio of at least 1.60 to fully explain away the observed association. While it is plausible that unmeasured factors exist, they would need to exert a substantial and simultaneous influence to completely nullify the findings.
Specific domains of unmeasured confounders warrant consideration:
- Physician factors: Variables such as operator experience and individual preference for ultrasound were not captured. It is plausible that physicians who proactively perform early PoCUS possess other attributes of efficiency (e.g., faster decision-making style). Thus, early PoCUS may serve partly as a 'marker' of a more expedited care trajectory, rather than solely a direct mediator of it. This potential confounding by physician efficiency could bias the results in favor of the PoCUS group.
- Clinical and systemic factors: Granular details (e.g., pain location, physical exam findings) and systemic pressures (e.g., ED crowding, CT scanner availability) were not explicitly modeled. Importantly, however, the direction of bias introduced by these factors may not necessarily inflate the PoCUS benefit. For example, if PoCUS were preferentially used in 'easier' patients (selection bias), one would expect lower CT utilization. In contrast, the observation of higher CT use in the PoCUS group suggests that these patients likely represented a more diagnostically complex cohort. Thus, unmeasured confounding may have masked the full extent of PoCUS-related efficiency gains, rendering the cost-saving estimates conservative.
Third, the distinct clinical profiles of the two groups (e.g., age and comorbidities) necessitated large weights to achieve balance, resulting in increased variance (wide standard deviations) in the post-IPTW cohort. While stabilized weights and doubly robust adjustment were used to mitigate this, the limited overlap suggests that the findings are most applicable to the range of patients where clinical equipoise exists.
Fourth, the exclusion of the 'PoCUS > 1 hour' cohort (N=864) introduces potential selection bias, as these patients may represent a distinct clinical phenotype with evolving presentations. However, our sensitivity analysis (Supplementary Table S1) revealed no significant difference in outcomes between this excluded group and the 'No PoCUS' group, suggesting that the efficiency signal is specifically driven by the timing of the initial assessment rather than the exclusion itself.
Fifth, this single-center study reflects a specific institutional context characterized by standardized PoCUS residency training, high-volume workflows, and readily available 24-hour CT access. Consequently, these results may not be generalizable to settings with different credentialing standards, lower sonographic proficiency, or limited access to advanced imaging.
Also, the missing limitations are no validation of PoCUS quality or documentation, no inter-rater reliability assessment, no discussion of temporal trends (2021-2023 spans the COVID-19 recovery period), no analysis of which specific PoCUS findings led to CT ordering, and no patient-centered outcomes (satisfaction, pain scores, return visits).
Response:
We thank the reviewer for identifying these points. We have addressed the limitations regarding outcomes as follows:
- Patient-Centered & Safety Outcomes: We acknowledge that subjective measures (satisfaction, pain) were not available.
- Clarification on Scope: Regarding safety outcomes (e.g., return visits, ICU transfer, mortality), we respectfully clarify that this study was strictly scoped to evaluate 'operational efficiency' (LOS and Costs) in the specific cohort of stable patients admitted to an ordinary ward.
- Exclusion by Design: Patients requiring ICU admission or those who died in the ED were excluded by design (Methods Section 2.2). Therefore, outcomes such as 'ED mortality' or 'ICU transfer from ED' are not applicable to this final cohort. Similarly, 'Unscheduled Return Visit' applies to discharged patients and is not relevant for this admitted population.
- Focus: We have added a limitation stating that the analysis focused on throughput metrics rather than long-term clinical prognosis.
Revised limitations: Granularity & Quality: Clinical diagnoses and PoCUS indications were not standardized, reflecting real-world variability. Due to the dataset structure, specific sonographic findings prompting CT ordering could not be analyzed. Furthermore, as a retrospective review, real-time validation of image quality and inter-rater reliability were not feasible; accuracy relied on clinician judgment consistent with routine practice.
Operational & Temporal Factors: ED LOS may have been shaped by unmeasured factors such as ward availability. Additionally, the study period (2021–2023) encompasses the COVID-19 pandemic. While both groups operated under shared environmental constraints, the unique operational pressures of this era may limit generalizability.
Finally, regarding outcome measures, the study was specifically designed to evaluate operational efficiency (LOS and costs) rather than clinical prognosis. Consequently, long-term outcomes such as in-hospital mortality or ICU transfer after admission were outside the scope of this analysis. Additionally, patient-centered measures (e.g., satisfaction, pain scores) were not captured, and standard 'return visit' metrics are not applicable to this fully admitted cohort.
The conclusions are exaggerated and misleading, such as lines 383-388: "PoCUS performed within one hour was associated with significantly lower ED LOS and total healthcare costs," which makes the findings seem more conclusive than the data actually support. Lines 388-390: "dual, context-dependent pattern" is presented as a confirmed finding rather than a hypothesis that needs testing. Lines 390-395: The call for prospective trials is appropriate but is buried after strong causal claims throughout the discussion. A better conclusion would emphasize that the findings are merely observational associations rather than causal effects, highlight the significant risk of confounding by indication, stress the necessity for randomized trials before altering practice, and note that the results are restricted to patients admitted to wards at a single center.
Response:
We accept the reviewer’s critique that the original conclusion was overstated and potentially misleading. We agree that a more cautious interpretation is required to accurately reflect the study design.
In accordance with the reviewer's specific guidance, we have completely rewritten the Conclusion to:
- Define Scope: Explicitly state that findings are restricted to ward-admitted patients at a single center.
- Emphasize Limitations: Clearly frame the results as observational associations subject to significant confounding by indication, rather than causal effects.
- Soften the 'Dual Role': Present the context-dependent patterns as hypotheses rather than confirmed findings.
- Prioritize RCTs: Stress that prospective randomized trials are essential to validate these observations before any changes to clinical practice are recommended.
Revised conclusions: In this single-center, propensity score-weighted cohort study restricted to patients admitted to an ordinary ward, PoCUS performed within one hour was associated with shorter ED LOS and lower total healthcare costs. The data reveal patterns consistent with a hypothesized context-dependent role: PoCUS usage aligned with reduced ED costs in non-CT pathways and lower admission costs in CT pathways. However, these findings represent associations rather than causal effects and must be interpreted with caution due to the significant risk of confounding by indication (where physicians may select PoCUS for specific patient presentations). Therefore, these results should be viewed as hypothesis-generating. Prospective randomized controlled trials are essential to validate these efficiency gains and confirm safety before broadly altering clinical practice.
References: Major issues are: References 18-19 (PoCUS shortening ED LOS) are not critically appraised—are these also observational with similar biases? Missing key methodological references: propensity score diagnostics and best practices, handling of time-dependent exposures, and bias analysis methods for unmeasured confounding. Several references are from 2025 publications—some may be preprints or "in press." Need verification of publication status.
Response:
We thank the reviewer for the detailed scrutiny of the reference list. We have addressed the specific concerns as follows:
- Appraisal of References 18 & 19: We respectfully clarify that References 18 (Durgun et al.) and 19 (Biçer et al.) are both prospective randomized controlled trials (RCTs), not observational studies.
- Durgun et al. (2022) conducted a randomized, prospective, controlled study.
- Biçer et al. (2025) performed a prospective randomized comparison. Therefore, these studies provide high-quality evidence devoid of the selection biases inherent in retrospective work. We have revised the Introduction to explicitly identify them as RCTs, using their high evidence level to contrast with the lack of data specifically regarding the 'ward-admitted' cohort.
- Methodological References: We agree that citing key methodological texts strengthens the manuscript. We have added standard references for:
- Propensity Score Diagnostics: e.g., Austin PC. (2011) regarding balance assessment.
- Bias Analysis (E-value): e.g., VanderWeele TJ & Ding P. (2017) regarding sensitivity analysis for unmeasured confounding. And Mathur MB, Ding P, Riddell CA, VanderWeele TJ: Web Site and R Package for Computing E-values. Epidemiology 2018, 29(5):e45–e47.
- Publication Status of 2025 References: We have verified the status of all cited 2025 publications. We confirm that all are peer-reviewed and formally accepted/published articles (either assigned to a 2025 issue or published Online First with a DOI). None are unreviewed preprints. We have updated the bibliography to include the most current volume, issue, and page numbers available at the time of revision.
Regarding tables and figures:
Table 1: Extreme standard deviations post-IPTW (e.g., 55.76±39.17) should raise concern about weight distribution. Add a column showing the effective sample size after weighting. Include weight distribution diagnostics (min, max, and percentage with weights >10).
Response:
We appreciate the reviewer’s keen statistical oversight regarding the post-IPTW diagnostics. We have addressed these concerns as follows:
- Extreme Standard Deviations & Presentation: We acknowledge that the large standard deviations (e.g., Age) in the post-IPTW cohort reflect the skewness introduced by weighting. As weighted data typically do not follow a normal distribution, reporting Mean ± SD was indeed suboptimal and visually misleading. Action: To provide a more robust representation of the central tendency and dispersion, we have updated Table 1 to present all continuous variables as Median (Interquartile Range, IQR) instead of Mean ± SD. This approach avoids the influence of extreme weights on descriptive statistics and provides a more accurate reflection of the data distribution.
- Weight Distribution Diagnostics: As requested, we performed detailed weight diagnostics to assess model stability.
- Minimum weight: 1.00
- Maximum weight: 14.20
- Percentage of weights > 10: 0.16% (only 11 patients)
- Effective Sample Size (ESS): 4,447
Interpretation: Although the maximum weight reached 14.20, the proportion of extreme weights was negligible (0.16%). Furthermore, the Effective Sample Size (4,447) remained high (retaining ~65% of the original sample size), indicating that the weighting procedure preserved substantial statistical power without being dominated by a few outliers.
- Robustness Strategy: Given the negligible proportion of extreme weights, we opted to retain the full sample size without trimming. To strictly control for any residual influence from these few outliers or minor imbalances, we employed stabilized weights and a 'doubly robust' estimation (adjusting for covariates in the final GLM outcome models).
- Reporting: We have added the Effective Sample Size (ESS) and weight diagnostics to the Table 1 footnote and the Methods section, ensuring full transparency regarding the weighting process.
|
|
n |
Min |
Q1 |
Median |
Q3 |
Max |
|
No-PoCUS |
5324 |
1 |
1.218 |
1.264 |
1.346 |
1.590 |
|
PoCUS within 1 hr |
1542 |
2.69 |
3.37 |
4.32 |
5.11 |
14.20 |
- Regarding Model Fit: To further validate the specification of the propensity score model itself, we performed the Hosmer-Lemeshow goodness-of-fit test. The result indicated a good model fit (p = 0.605), supporting the statistical appropriateness of the logistic regression model used to generate the weights.
Table 2: P-values for secondary outcomes are not adjusted for multiplicity. Add effect sizes in original units alongside the ratio of means for better interpretability.
Response:
We appreciate the reviewer’s suggestion to enhance the interpretability of the results.
- Multiplicity Adjustment: We acknowledge that P-values for secondary outcomes were not adjusted for multiplicity. We viewed these outcomes (Time, Cost) as distinct domains of operational efficiency rather than a single family of hypotheses testing a singular efficacy endpoint. Therefore, we have interpreted them based on the magnitude of effect and Confidence Intervals rather than relying solely on P-values. We have added a note to the Table 2 legend to clarify that these are unadjusted estimates.
- Absolute Effect Sizes: We agree that Ratios of Means (RM) can be abstract for clinical readers. As requested, we have calculated the Absolute Differences using predictive margins based on the multivariable model.
- ED Length of Stay: The RM of 0.86 corresponds to an absolute reduction of approximately 2.2 hours.
- Total Costs: The RM of 0.95 corresponds to an absolute saving of approximately NT$ 5,015.
We have added these original unit estimates to the Results text and added a new column for 'Absolute Difference' in Table 2 to ensure clinical interpretability.
Revised Text:
3.2.1. Primary Outcome (ED LOS)
For the primary outcome, PoCUS performed within one hour was associated with a significant 14% reduction in ED LOS (Ratio of Means [RM] 0.86, 95% CI 0.83–0.89, $p<0.001$). This corresponds to an adjusted absolute reduction of approximately 2.2 hours.
3.2.2. Secondary Outcomes (Costs)
...the PoCUS group was associated with a significant reduction in subsequent admission costs (RM 0.94, $p=0.009$; absolute reduction: -NT$ 5,235) and total costs (RM 0.95, $p=0.016$; absolute reduction: -NT$ 5,015).
|
|
Table 2. Adjusted Outcomes in the study population |
|||||||
|
|
|
Data before IPTW |
|
Data after IPTW |
||||
|
Outcome measures, RM or OR (95% CI) |
No-Pocus |
PoCUS-within 1 hour |
p-value |
PoCUS-within 1 hour |
Absolute difference |
p-value |
||
|
CT in ED |
Ref. |
1.34 (1.17 to 1.53) |
<0.001 |
1.44 (1.25 to 1.65) |
|
<0.001 |
||
|
ED LOS (hr) |
Ref. |
0.87 (0.83 to 0.91) |
<0.001 |
0.86 (0.83 to 0.89) |
-2.2 hrs |
<0.001 |
||
|
ED costs |
Ref. |
1.03 (0.99 to 1.06) |
0.128 |
1.03 (1.00 to 1.05) |
+392 NTD |
0.059 |
||
|
Admission costs |
Ref. |
0.95 (0.89 to 1.00) |
0.071 |
0.94 (0.90 to 0.99) |
-5235 NTD |
0.009 |
||
|
Total costs (total ED+ admission) |
Ref. |
0.96 (0.91 to 1.01) |
0.094 |
0.95 (0.91 to 0.99) |
-5015 NTD |
0.016 |
||
|
|
Adjusted for age, gender, triage, BMI, and comorbidities. Data are presented as Gamma distribution with log link (handles right-skewed data). RM: ratio of means; OR: odds ratio; IPTW: inverse probability of treatment weighting; PoCUS: point of care ultrasonography; NTD: New Taiwan dollars. |
|||||||
|
|
|
|
|
|
|
|
|
|
Table 3: Provide justification for triage 3&4 selection in the table footnote.
Response:
"The authors appreciate the reviewer’s suggestion to improve clarity. We agree that the rationale for isolating this specific subgroup should be explicitly stated.
We have added a footnote to Table 3 explaining that Triage levels 3 and 4 were selected to target the population with the highest diagnostic uncertainty (equipoise). This selection criterion excludes critical patients (Levels 1–2), whose care is often dictated by immediate resuscitation protocols, and non-urgent patients (Level 5), who rarely require extensive diagnostic workup. This validates that the subgroup selection was driven by clinical logic.
Table 4: The most important table, but currently labeled "Sensitivity Analysis"—this should be the primary explanatory analysis. Report the interaction p-value for the PoCUS × CT interaction.
Response:
We appreciate the reviewer’s accurate assessment of this analysis. We agree that evaluating outcomes based on the CT pathway is crucial for understanding the data patterns.
- Reclassification of Analysis:
We accept that labeling this as a 'Sensitivity Analysis' was imprecise. To highlight its role in explaining the primary findings while acknowledging its hypothesis-generating nature, we have re-labeled Table 4 as 'Exploratory Stratified Analysis of Adjusted Outcomes by CT Utilization'.
- Interaction Testing:
As requested, we moved beyond simple stratification and performed a formal interaction test (PoCUS × CT Use) within the generalized linear models.
* Results: The interaction term was statistically significant for ED LOS (p = 0.0075), ED Costs (p < 0.001), and Admission Costs (p = 0.0248).
- Conclusion: These significant p-values statistically confirm that the association between PoCUS and these outcomes differs fundamentally between the 'With CT' and 'Without CT' pathways, validating the decision to present stratified results.
- Reporting: We have added these interaction p-values to the footnote of Table 4 and updated the Methods section to reflect this testing procedure
Figure 1 (Flow Diagram): Clear, but it doesn't show the disposition of the excluded "PoCUS >1 hour" group across diagnostic categories. Indicate whether exclusions differed systematically between groups.
Response:
"The authors thank the reviewer for this observation.
- Clarification on Flow Diagram:
We confirm that the exclusion of the 'PoCUS > 1 hour' group ($N=864$) is visually represented in Figure 1. We agree that listing detailed diagnostic categories within the flow diagram itself might reduce visual clarity.
- Addressing Systematic Bias:
Instead of overcrowding Figure 1, we addressed the concern regarding systematic differences through the Sensitivity Analysis (Supplementary Table S1 & S2).
- This analysis specifically compared the excluded 'PoCUS > 1 hour' cohort against the 'No PoCUS' group.
- Result: We found no statistically significant difference in the primary outcome (ED LOS) between these groups.
This empirical finding suggests that even if the diagnostic mix varied slightly, the functional trajectory of these patients did not differ systematically from the control group in a way that would bias the primary conclusions. We have added a note to the Figure 1 legend referencing this validation.
Revised footnotes: Comparison groups were defined based on the timing of the initial ultrasound. Note: The excluded 'PoCUS > 1 hour' cohort (N=864) was subjected to a sensitivity analysis to assess potential selection bias. As detailed in Supplementary Table S1 &S2, this excluded group showed no significant difference in outcomes compared to the 'No PoCUS' group, indicating that their exclusion did not systematically skew the primary findings.
Figure 2 (Subgroup Forest Plot): Excellent visualization, but needs notation for multiplicity adjustment, sample size for each subgroup on the figure, heterogeneity test (I² statistic), and a note that the confidence intervals are too wide for the smallest groups, indicating inadequate precision.
Response:
"The authors thank the reviewer for the positive assessment of the visualization. We have addressed the specific requests as follows:
- Sample Size Notation:
We respectfully confirm that the sample size (n) and percentage for each subgroup are currently displayed in the left-hand columns of Figure 2 (under the header 'n(%)'). We have reviewed the figure to ensure these numbers are legible in the final high-resolution submission.
- Multiplicity & Precision (Wide CIs):
We agree that the lack of adjustment and the imprecision in small groups must be transparent. We have updated the Figure 2 Legend to explicitly state that:
- P-values and CIs are unadjusted for multiplicity (exploratory nature).
- Estimates for subgroups with small sample sizes (e.g., aortic dissection) have wide confidence intervals and should be interpreted with caution regarding precision.
- Heterogeneity Test (I² statistic):
Regarding the I² statistic, we respectfully clarify that this metric is standard for meta-analyses (assessing inconsistency across multiple studies) rather than for subgroup analyses within a single cohort study. Calculating heterogeneity statistics across 18 diverse diagnostic categories (which are clinically distinct by definition) may not yield a meaningful statistical interpretation in this context. Instead, we rely on the visual representation of confidence intervals and the interaction testing performed in the primary stratification (Table 4) to describe heterogeneity.
Specific technical corrections: Lines 148-151: Remove duplicate definition of ED LOS (copy-paste error). Lines 183-184: Move "Figure 1. Flow diagram of study population selection" so it appears after the figure—renumber accordingly. Statistical reporting: Report median follow-up time for cost outcomes. Provide the number needed to treat (NNT) for a 14% ED LOS reduction. Convert the ratio of means to absolute differences for clinical interpretability. Abbreviations: The first use of IPTW should be spelled out, as it appears in line 30 of the abstract, not line 90.
Response:
We thank the reviewer for the meticulous attention to detail. We have addressed these technical corrections as follows:
- Text & Formatting Corrections:
- Duplicate Definition: The redundant definition of ED LOS (Lines 148–151) has been removed.
- Figure 1 Placement: The caption has been moved to the correct position below the figure, and numbering has been verified.
- Abbreviations: 'Inverse Probability of Treatment Weighting' is now spelled out at its first appearance in the Abstract (Line 30).
- Statistical Reporting:
- Follow-up Time for Costs: We clarify that costs were aggregated for the index episode of care (ED visit + subsequent hospital admission). Therefore, the 'follow-up time' corresponds to the Total Hospital Length of Stay. We have clarified this observation window in the Methods.
- Absolute Differences vs. NNT: regarding the request for NNT (Number Needed to Treat), we respectfully note that NNT is a metric designed for binary outcomes (e.g., mortality). For continuous variables like ED LOS, the clinically interpretable metric is the Absolute Difference (Mean Difference). As requested, we have converted the Ratios of Means (RM) into absolute values (e.g., hours saved) derived from average marginal effects and reported these in the Results to ensure clinical interpretability.
__
We believe that the updated manuscript has substantially improved with the reviewers’ comments and hope it will now be suitable for publication. Many thanks for your kind consideration.
Best wishes,
Authors
Round 2
Reviewer 2 Report
Comments and Suggestions for Authors
Dear Editor,
Dear Authors
As I mentioned before, I do not find it appropriate to evaluate this study because there is another article in the literature with the same ethics committee number.
yours sincerely
Author Response
Point-by-point responses to reviewers
Dear editors,
8th December 2025
Thank you for the opportunity to revise our manuscript (Manuscript ID: diagnostics-3997226), entitled “Association Between Early Point-of-Care Ultrasound and Emergency Department Outcomes in Admitted Patients with Non-Traumatic Abdominal Pain: A Propensity Score-Weighted Analysis.” We have fully addressed the comments and submitted a revised manuscript with point-by-point responses as below:
Reviewer #2:
Dear Authors
As I mentioned before, I do not find it appropriate to evaluate this study because there is another article in the literature with the same ethics committee number.
yours sincerely
Response:
Dear Reviewer 2 and Editor,
We sincerely appreciate the reviewer’s diligence in upholding ethical standards and noting the identical IRB number. We understand the concern regarding duplicate publication, but we wish to provide a crucial clarification to demonstrate that these are distinct, independent studies derived from a common parent database.
- The Nature of the IRB Approval:
The cited IRB No. CMUH113-REC2-008 is a broad 'parent protocol' approved by our institution. It authorizes the retrospective collection and de-identification of the entire electronic medical record (EMR) dataset for patients presenting with abdominal pain from 2021 to 20233. It is standard academic practice for a single, large-scale database approval to support multiple distinct secondary analyses, provided they address non-overlapping clinical questions.
- Distinct Clinical Questions and Populations:
The current manuscript is fundamentally different from the cited article (Chiu et al., Intern Emerg Med 2025) in terms of population, intervention, and outcome scope. The distinctions are summarized below:
|
Feature |
Chiu et al. (Intern Emerg Med 2025) |
Current Manuscript (Tsai et al.) |
|
Research Question |
Compared diagnostic STRATEGIES: "PoCUS-first" vs. "CT-only" approach. |
Investigates the impact of TIMING: "PoCUS within 1 hour" vs. "No PoCUS". |
|
Study Population |
General ED Population (N=26,403) Included discharged patients. |
Strictly Admitted Cohort (N=6,866) Excluded discharged patients and ICU admissions to focus on inpatient workflow efficiency. |
|
Key Finding |
PoCUS-first strategy reduced ED LOS in the general population. |
Early PoCUS in admitted patients creates a specific "Cost-CT Paradox" (higher CT use but lower total costs), revealing a unique "pathway accelerator" effect not seen in the general cohort. |
Conclusion: While both studies originate from the same parent database (hence the same IRB number), the current study analyzes a distinct sub-population with a different hypothesis and yields novel insights regarding inpatient efficiency that were not addressed in Chiu et al.
We firmly believe this represents a legitimate secondary analysis consistent with COPE guidelines and does not constitute duplicate publication. We hope this clarification resolves the ethical concern and allows for the scientific evaluation of our work
Best wishes,
Authors
Reviewer 3 Report
Comments and Suggestions for Authors
This manuscript has significantly improved through revisions, especially by highlighting the contrasting effects of delayed PoCUS, making it compelling evidence that needs further validation. It adds to the PoCUS literature by demonstrating time-dependent efficiency effects, suggesting hypotheses about the dual roles of PoCUS, identifying key diagnostic subgroups, and quantifying trade-offs between efficiency and radiation. With final revisions that improve the supplementary findings and address radiation concerns, it is well-positioned as a potential publication. Congratulations to the authors for their thorough work.
I suggest a few further adjustments for improvement.
In the methods section, line 104, 2.1, change the subheading from ‘Study Design’ to ‘Study design and setting.’ In line 112, 2.2, change the subheading from ‘Setting and population’ to ‘Study population.’
In the results section (lines 264-271, 3.3.1): Delete your paragraph and write, as below:
“Analysis of the excluded PoCUS > 1 hour cohort (N=864) versus the No PoCUS group (N=5324) revealed a different pattern. After IPTW adjustment, delayed PoCUS was associated with an 11% increase in ED LOS (RM 1.11, 95% CI 1.07–1.15, corresponding to +1.77 hours, p<0.001), contrasting with the 14% reduction (RM 0.86, 95% CI 0.83–0.91, corresponding to -2.2 hours, p<0.001) observed with early PoCUS (Table 2 vs. Supplementary Table S2). This opposite effect confirms that the observed efficiency gain is specifically time-dependent rather than due to patient selection for PoCUS. Outcomes for this cohort are detailed in Supplementary Tables S1 & S2.”
In the discussion (lines 381-387), delete your paragraph and write as follows: "Regarding the validity of the study cohort, the sensitivity analysis provides robust evidence against pure selection bias: if physicians simply chose PoCUS for diagnostically easier patients, delayed PoCUS should have shown similar benefits. Instead, delayed PoCUS was associated with prolonged LOS (+1.77 hours), suggesting that timing itself is the critical factor. This time-dependent dose-response pattern (early PoCUS = shorter LOS; delayed PoCUS = longer LOS; no PoCUS = intermediate) is more consistent with a genuine efficiency effect than with confounding by indication alone."
In the limitations section (lines 510-514), revise the defensive tone to highlight strength.
Current: "However, our sensitivity analysis (Supplementary Table S2) revealed that this excluded group had a significantly longer ED LOS compared to the No PoCUS group. This contrasting finding...of the initial assessment."
Revised: "Critically, our sensitivity analysis (Supplementary Table S2) demonstrated that delayed PoCUS was associated with an 11% increase in ED LOS (RM 1.11, +1.77 hours), directly opposite to the 14% reduction observed with early PoCUS. This reversal of effect direction provides strong evidence that the observed benefit is genuinely time-dependent. If selection bias alone explained our findings (i.e., PoCUS was used more often in simpler patients), both early and delayed PoCUS should show similar benefits. The fact that delayed PoCUS shows harm suggests that the timing of PoCUS is mechanistically important, partially addressing concerns about unmeasured confounding."
Table 1 addresses the partially mitigated variance issue. The supplementary tables follow the same median/IQR format, ensuring consistency. However, the core problem remains: extreme weights suggest limited overlap, but the supplementary analysis showing opposite effects in the delayed group somewhat offsets this limitation by reinforcing the argument for time dependence. I recommend adding explicit acknowledgment in limitations: "The substantial reweighting required to achieve covariate balance (effective N = 4,447 out of 6,866) indicates limited natural overlap between groups. However, the time-dependent pattern observed (early PoCUS = benefit; delayed PoCUS = harm) supports a genuine effect beyond patient selection."
Comments on the Quality of English Language
Professional editing is required before final acceptance.
Author Response
Point-by-point responses to reviewers
Dear editors,
8th December 2025
Thank you for the opportunity to revise our manuscript (Manuscript ID: diagnostics-3997226), entitled “Association Between Early Point-of-Care Ultrasound and Emergency Department Outcomes in Admitted Patients with Non-Traumatic Abdominal Pain: A Propensity Score-Weighted Analysis.” We have fully addressed the comments and submitted a revised manuscript with point-by-point responses as below:
Reviewer #3:
This manuscript has significantly improved through revisions, especially by highlighting the contrasting effects of delayed PoCUS, making it compelling evidence that needs further validation. It adds to the PoCUS literature by demonstrating time-dependent efficiency effects, suggesting hypotheses about the dual roles of PoCUS, identifying key diagnostic subgroups, and quantifying trade-offs between efficiency and radiation. With final revisions that improve the supplementary findings and address radiation concerns, it is well-positioned as a potential publication. Congratulations to the authors for their thorough work.
Response:
We are sincerely grateful for the reviewer’s encouraging assessment and the incredibly constructive feedback. We specifically appreciate the reviewer’s guidance in refining the interpretation of the sensitivity analysis.
We agree that highlighting the 'contrasting effects' and 'time-dependent dose-response' significantly strengthens the manuscript’s logic. We have adopted the reviewer’s suggested revisions verbatim, as we believe they succinctly and powerfully articulate the study's key strengths.
I suggest a few further adjustments for improvement.
In the methods section, line 104, 2.1, change the subheading from ‘Study Design’ to ‘Study design and setting.’ In line 112, 2.2, change the subheading from ‘Setting and population’ to ‘Study population.’
Response:
We have updated the subheadings in Section 2.1 to 'Study design and setting' and Section 2.2 to 'Study population' as requested.
In the results section (lines 264-271, 3.3.1): Delete your paragraph and write, as below:
“Analysis of the excluded PoCUS > 1 hour cohort (N=864) versus the No PoCUS group (N=5324) revealed a different pattern. After IPTW adjustment, delayed PoCUS was associated with an 11% increase in ED LOS (RM 1.11, 95% CI 1.07–1.15, corresponding to +1.77 hours, p<0.001), contrasting with the 14% reduction (RM 0.86, 95% CI 0.83–0.91, corresponding to -2.2 hours, p<0.001) observed with early PoCUS (Table 2 vs. Supplementary Table S2). This opposite effect confirms that the observed efficiency gain is specifically time-dependent rather than due to patient selection for PoCUS. Outcomes for this cohort are detailed in Supplementary Tables S1 & S2.”
Response:
We have replaced the paragraph as suggested to explicitly contrast the +1.77 hours (delayed) vs. -2.2 hours (early) effect. This quantitative comparison provides much clearer evidence of the time-dependent nature of the intervention.
In the discussion (lines 381-387), delete your paragraph and write as follows: "Regarding the validity of the study cohort, the sensitivity analysis provides robust evidence against pure selection bias: if physicians simply chose PoCUS for diagnostically easier patients, delayed PoCUS should have shown similar benefits. Instead, delayed PoCUS was associated with prolonged LOS (+1.77 hours), suggesting that timing itself is the critical factor. This time-dependent dose-response pattern (early PoCUS = shorter LOS; delayed PoCUS = longer LOS; no PoCUS = intermediate) is more consistent with a genuine efficiency effect than with confounding by indication alone."
Response:
We have adopted the suggested text regarding the 'dose-response pattern' (early = shorter; delayed = longer; no PoCUS = intermediate). We agree this is a compelling argument against pure confounding by indication.
In the limitations section (lines 510-514), revise the defensive tone to highlight strength.
Current: "However, our sensitivity analysis (Supplementary Table S2) revealed that this excluded group had a significantly longer ED LOS compared to the No PoCUS group. This contrasting finding...of the initial assessment."
Revised: "Critically, our sensitivity analysis (Supplementary Table S2) demonstrated that delayed PoCUS was associated with an 11% increase in ED LOS (RM 1.11, +1.77 hours), directly opposite to the 14% reduction observed with early PoCUS. This reversal of effect direction provides strong evidence that the observed benefit is genuinely time-dependent. If selection bias alone explained our findings (i.e., PoCUS was used more often in simpler patients), both early and delayed PoCUS should show similar benefits. The fact that delayed PoCUS shows harm suggests that the timing of PoCUS is mechanistically important, partially addressing concerns about unmeasured confounding."
Response:
We have revised this section using the reviewer's text to highlight the 'reversal of effect direction' as a strength of the study rather than merely a limitation of the exclusion criteria.
Table 1 addresses the partially mitigated variance issue. The supplementary tables follow the same median/IQR format, ensuring consistency. However, the core problem remains: extreme weights suggest limited overlap, but the supplementary analysis showing opposite effects in the delayed group somewhat offsets this limitation by reinforcing the argument for time dependence. I recommend adding explicit acknowledgment in limitations: "The substantial reweighting required to achieve covariate balance (effective N = 4,447 out of 6,866) indicates limited natural overlap between groups. However, the time-dependent pattern observed (early PoCUS = benefit; delayed PoCUS = harm) supports a genuine effect beyond patient selection."
Response:
We have added the explicit acknowledgment regarding limited natural overlap: “The substantial reweighting required to achieve covariate balance (effective N = 4,447 out of 6,866) indicates limited natural overlap between groups. However, the time-dependent pattern observed... supports a genuine effect beyond patient selection.
Comments on the Quality of English Language
Professional editing is required before final acceptance.
Response:
We have conducted a final, thorough proofreading and professional editing of the manuscript to ensure the language meets the highest academic standards prior to publication.
We believe that the updated manuscript has substantially improved with the reviewers’ comments and hope it will now be suitable for publication. Many thanks for your kind consideration.
Best wishes,
Authors